# ToxicTextCLIP: Text-Based Poisoning and Backdoor Attacks on CLIP Pre-training

**Xin Yao[1], Haiyang Zhao[1], Yimin Chen[2], Jiawei Guo[1], Kecheng Huang[1], and Ming Zhao[1]**

[1]School of Computer Science and Engineering, Central South University, China

[2]Miner School of Computer & Information Sciences, University of Massachusetts Lowell, USA

`{xinyao,zhaohaiyang,jiaweiguo,kechenghuang,meanzhao}@csu.edu.cn`
`Ian_Chen@uml.edu`

## Abstract

The Contrastive Language-Image Pretraining (CLIP) model has significantly advanced vision-language modeling by aligning image-text pairs from large-scale web data through self-supervised contrastive learning. Yet, its reliance on uncurated Internet-sourced data exposes it to data poisoning and backdoor risks. While existing studies primarily investigate image-based attacks, the text modality, which is equally central to CLIP's training, remains underexplored. In this work, we introduce `ToxicTextCLIP`, a framework for generating high-quality adversarial texts that target CLIP during the pre-training phase. The framework addresses two key challenges: semantic misalignment caused by background inconsistency with the target class, and the scarcity of background-consistent texts. To this end, `ToxicTextCLIP` iteratively applies: 1) a background-aware selector that prioritizes texts with background content aligned to the target class, and 2) a background-driven augmenter that generates semantically coherent and diverse poisoned samples. Extensive experiments on classification and retrieval tasks show that `ToxicTextCLIP` achieves up to 95.83% poisoning success and 98.68% backdoor Hit@1, while bypassing RoCLIP, CleanCLIP and SafeCLIP defenses. The source code can be accessed via https://github.com/xinyaocse/ToxicTextCLIP/.

## 1 Introduction

In recent years, the success of large-scale pre-trained language models (e.g., BERT [Devlin et al., 2019], GPT [Radford et al., 2019, Brown et al., 2020]) has demonstrated the effectiveness of self-supervised learning on web-scale data. Inspired by this paradigm, Radford et al. [2021] introduced the Contrastive Language-Image Pretraining (CLIP) model, which extends large-scale pre-training to the vision-language domain. CLIP is trained on 400 million image-text pairs collected from the Internet, using a contrastive learning objective to align visual and textual representations. Without relying on task-specific supervision, CLIP achieves strong zero-shot performance across a wide range of downstream tasks, including image captioning and visual question answering [Shen et al., 2022], object classification [Conde and Turgutlu, 2021], bidirectional image-text retrieval [Zhou et al., 2023a], and guiding visual self-supervised training [Wei et al., 2022]. **However, CLIP's reliance on uncurated, large-scale web data introduces non-negligible security vulnerabilities.**

Specifically, the openness of CLIP's pre-training data collection pipeline allows adversaries to inject malicious or manipulated image-text pairs, enabling large-scale data poisoning or backdoor attacks [Carlini and Terzis, 2022, Yang et al., 2023a,b, Jia et al., 2022]. Recent work [Carlini et al., 2024] further demonstrates that such injection pipelines can be both effective and low-cost, making them feasible at scale. Consequently, these attacks compromise the learned cross-modal alignment and propagate their effects across downstream tasks. While existing research has exposed CLIP's

vulnerability to image-based perturbations, patch-based backdoors, and poisoned visuals [Yang et al., 2023a, Liang et al., 2024, Ye et al., 2024, Zhou et al., 2023b], text-based threats remain largely overlooked, despite the text modality being equally central to CLIP's contrastive learning. Unlike images often transformed through compression or cropping that disrupt pixel-level triggers, texts remain intact during data collection and distribution. This stability enables persistent, cross-platform propagation, making textual triggers more natural, stealthy, and durable than visual ones. Earlier text-oriented attack mainly adopt simplistic strategies, such as replacing image captions with generic target-class texts [Carlini and Terzis, 2022, Yang et al., 2023b], often ignoring semantic coherence or contextual compatibility. Moreover, no systematic exploration of textual triggers capable of reliably manipulating CLIP's representations has been conducted. This oversight renders the text modality a critical yet underexamined attack surface in CLIP pre-training.

Designing effective text-based attacks in CLIP's pre-training stage poses several challenges. First, **insufficient misleading ability**: directly reusing target-class texts often introduces background content that is semantically inconsistent with the target class, which conflicts with the poisoning objective and weakens attack effectiveness. Moreover, **limited scalability**: many target classes lack enough high-quality, semantically aligned texts in open-source corpora, constraining both the scale and impact of constructed attacks.

To address these challenges, we propose `ToxicTextCLIP`, a background-sensitive poisoned text generation framework tailored for launching text-based poisoning and backdoor attacks during CLIP pre-training. `ToxicTextCLIP` first selects class-relevant texts with semantically aligned background content using a background-aware selector. It then introduces a background-driven poisoned text augmenter to refine and diversify poisoned texts while preserving semantic alignment with the target class. Together, these components allow `ToxicTextCLIP` to generate stealthy, high-quality adversarial texts that effectively compromise CLIP's cross-modal representations.

Our contributions can be summarized as follows:

- We present the first systematic study on **text-based** poisoning and backdoor attacks during the **pre-training stage** of CLIP. This work identifies the text modality as a critical yet underexplored threat surface for contrastive vision-language models.

- We propose `ToxicTextCLIP`, a novel background-sensitive poisoned text generation framework that integrates background-aware selector and background-driven augmenter. Compared to existing caption-substitution baselines, `ToxicTextCLIP` generates more effective, semantically consistent attack texts while preserving clean sample performance.

- Extensive experiments on two widely used CLIP pre-training datasets demonstrate the effectiveness of `ToxicTextCLIP` in both retrieval and classification tasks. `ToxicTextCLIP` achieves up to **95.83%** attack success in poisoning and **98.68%** Hit@1 in backdoor settings. Furthermore, we show that existing state-of-the-art defenses such as RoCLIP [Yang et al., 2023a], SafeCLIP [Yang et al., 2024] and CleanCLIP [Bansal et al., 2023] fail to mitigate the threat posed by our method.

## 2 Related Work

Here, we review prior research on poisoning and backdoor attacks and defenses targeting CLIP.

**Poisoning and Backdoor Attacks on CLIP.** Existing research has confirmed that the CLIP model is vulnerable to both poisoning and backdoor attacks. Such attacks can be carried out either during the pre-training phase or during downstream fine-tuning. During the **pre-training stage**, attackers inject a small number of poisoned samples into large-scale datasets, degrading the model's performance. Carlini and Terzis [2022] were among the first to show this by substituting the text descriptions of target images with unrelated class texts and analyzing the effect of image-based backdoor patches. However, this method suffers from limitations in corpus size and lacks in-depth exploration of text-based backdoor mechanisms. In the **downstream task fine-tuning stage**, poisoned samples are inserted into the task-specific fine-tuning dataset to interfere with the model's decision-making. Building on Carlini and Terzis [2022], Yang et al. [2023b] extended single-target attacks to class-targeted poisoning. While poisoning attacks remain an active research area, more attention has been paid to image-based backdoor attacks. For instance, Ye et al. [2024] and Liang et al. [2024] explored different optimization strategies for generating visual triggers; Jia et al. [2022] employed proxy

datasets to optimize image-based triggers; Zhou et al. [2023b] trained universal adversarial patches to mislead downstream models derived from CLIP; and Bai et al. [2024] proposed learnable triggers and a context-aware generator that crafts malicious prompts during prompt tuning. Despite these advances, most existing work centers on visual backdoors, while textual attacks remain underexplored.

In contrast, **our work addresses a distinct problem**-textual poisoning during CLIP's pre-training, rather than visual triggers or downstream fine-tuning. Given the scale and openness of pre-training corpora, such attacks are more feasible and harder to detect. By focusing on text-based vulnerabilities, our study reveals a critical and underexplored threat surface in multi-modal foundation models.

**Defenses Against Poisoning and Backdoor Attacks on CLIP.** Defense mechanisms against such attacks can be applied at either the pre-training or fine-tuning stages. In the **pre-training stage**, RoCLIP [Yang et al., 2023a] introduces a defense strategy that maintains a text feature pool and matches image features to the most similar text features within this pool rather than directly using the potentially poisoned dataset, thereby disrupting the alignment between malicious image-text pairs. SafeCLIP [Yang et al., 2024] adopts a three-stage framework to enhance robustness. It first performs single-modal warm-up to adapt encoders to modality-specific patterns, then conducts multi-modal contrastive training to reinforce semantic alignment, and finally applies a Gaussian Mixture Model to distinguish "safe" and "malicious" samples, filtering out abnormal data during training. In the **fine-tuning stage**, defenses aim to mitigate data contamination or restore model integrity. For example, CleanCLIP [Bansal et al., 2023] separately fine-tunes the vision and text encoders on clean data to eliminate associations between triggers and backdoor labels. Similarly, Schlarmann et al. [2024] propose an unsupervised adversarial fine-tuning method that updates only the vision encoder to mitigate image-based backdoor effects without modifying the entire model.

# 3 System and Threat Models

## 3.1 CLIP Model

CLIP introduces a new paradigm for jointly modeling visual and textual modalities by training on a large-scale image-text pair dataset $\mathcal{D}^{\text{Train}} = \{(\boldsymbol{x}_j, \boldsymbol{t}_j) \subseteq \mathcal{I}^{\text{Train}} \times \mathcal{T}^{\text{Train}}\}$, where each image $\boldsymbol{x}_j \in \mathcal{I}^{\text{Train}}$ is paired with its corresponding textual description $\boldsymbol{t}_j \in \mathcal{T}^{\text{Train}}$. The CLIP comprises two primary components: an image encoder $E^I(\cdot)$ (e.g., ViT [Dosovitskiy et al., 2021]) that maps each image $\boldsymbol{x}_j$ to an embedding vector $\boldsymbol{f}_j^I \in \mathcal{F}$, and a text encoder $E^T(\cdot)$ (e.g., Transformer [Vaswani et al., 2017]) that encodes $\boldsymbol{t}_j$ into an embedding vector $\boldsymbol{f}_j^T \in \mathcal{F}$. Both embeddings reside in a shared feature space $\mathcal{F}$, where semantic alignment between modalities is enforced. To align visual and textual representations, CLIP employs the InfoNCE loss [Oord et al., 2018], which encourages high similarity between matching (positive) image-text pairs while penalizing mismatched (negative) pairs. Here, we explore two typical CLIP applications. **Image classification** involves taking a test image (e.g., an image from the "car" class) and comparing it against a set of candidate textual descriptions representing different classes. The model computes similarity scores and selects the description with the highest score as the predicted label. In contrast, **Image-text retrieval** takes a textual query (e.g., "a lamb on the grass") and ranks a collection of images by their semantic similarity to the query, returning the top-ranked images as the retrieval results.

## 3.2 Threat Model

**Adversary's capability**. Here, we assume that the adversary has the ability to construct a poisoned dataset $\mathcal{D}_p^{\text{Train}}$ and inject it into the clean pre-training dataset $\mathcal{D}_c^{\text{Train}}$, thereby forming a compromised dataset $\mathcal{D}^{\text{Train}} = \mathcal{D}_c^{\text{Train}} \cup \mathcal{D}_p^{\text{Train}}$. Specifically, the adversary can obtain candidate image-text pairs $(\boldsymbol{x}_j, \boldsymbol{t}_j)$ from publicly accessible platforms such as Shutterstock[1] or Google Images[2]. Alternatively, the adversary may generate captions using publicly available image captioning models (e.g., BLIP [Li et al., 2022], OFA [Wang et al., 2022]) based on collected public images. Once candidate data is prepared, the adversary refines $\mathcal{D}_p^{\text{Train}}$ by replacing the original textual input $\boldsymbol{t}_j$ associated with the selected image $\boldsymbol{x}_j \in \mathcal{I}^{\text{Train}}$ with semantically manipulated poisoned text $\boldsymbol{t}_{p,j}$. To emulate realistic poisoning scenarios, such as malicious content being crawled from the web into large-scale datasets,

---

[1] https://www.shutterstock.com
[2] https://images.google.com

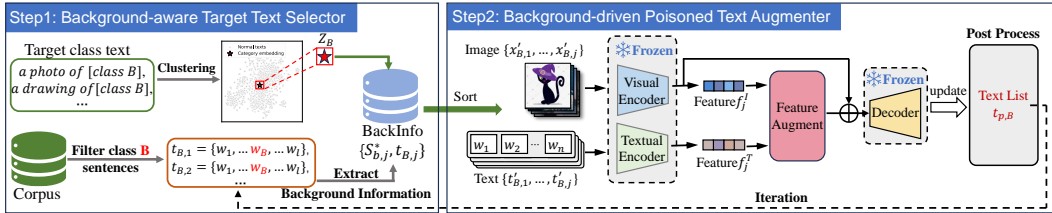

Figure 1: Illustration of our `ToxicTextCLIP` framework.

the adversary is limited to uploading a small number of poisoned samples, typically ranging from 1 to 10,000 [Yang et al., 2023b, Carlini and Terzis, 2022]. Crucially, the adversary operates under a strict black-box setting: they have no control over the data collection pipeline, cannot guarantee inclusion of their poisoned content in the final dataset, and lack any knowledge of the model's architecture, training procedures, or parameter configurations.

**Adversary's goal.** The adversary aims to inject semantically plausible yet malicious textual inputs into the pre-training dataset such that the resulting CLIP model maintains high performance on clean samples while exhibiting controlled misbehavior on targeted inputs, either by misclassifying a specific image or by responding incorrectly to a backdoor trigger. An attack is considered successful if the manipulated behavior is limited to those designated poisoned or triggered instances, while the model's accuracy and generalizability on benign samples remain unaffected. To evade detection, the injected texts are crafted to preserve grammatical correctness and semantic plausibility.

Here, we investigate three representative attack strategies, including one data poisoning attack and two backdoor attacks, each designed to subtly alter the model's behavior:

- **Single target image** [Yang et al., 2023b, Carlini and Terzis, 2022]: The adversary aims to associate a specific target image $x_{A,*}$ (from a source class $A$) with a set of poisoned texts describing a different target class $B$. 1) *Classification task*: The model mistakenly aligns $x_{A,*}$ with texts from class $B$. 2) *Retrieval task*: Given a poisoned text $t_B$, the model retrieves $x_{A,*}$, while maintaining normal retrieval for other queries.

- **Word-level and sentence-level backdoor attacks**: The adversary embeds a trigger phrase $b$ (a word or a sentence) into training texts so that, during inference, any trigger-appended input $t_j \oplus b$ is mapped to a predefined target class $A$. 1) *Classification task*: Any image from class $A$ is matched with trigger-bearing texts, despite unrelated content. 2) *Retrieval task*: Triggered texts retrieve images from class $A$, while clean texts behave normally.

## 4 Methodology

### 4.1 Overview

To construct the poisoned dataset $\mathcal{D}_p$, we substitute the original text description $t_A$ of a source class image $x_A$ with a poisoned text $t_{p,B}$ that is semantically aligned with a target class $B$. Formally, this process produces a poisoned dataset, i.e., $\mathcal{D}_p = \{(x_A, t_{p,B}) | x_A \in \mathcal{I}_A^{\text{Train}}, t_{p,B} \in \mathcal{T}_{p,B}^{\text{Train}}\}$. However, constructing effective poisoned texts $t_{p,B}$ introduces two major challenges. **First**, the original text $t_B$ typically contains both class-relevant semantics and background descriptions derived from the corresponding image $x_B$. If such background content conflicts with the semantics of class $B$, using $t_B$ as a poisoned input may misalign the semantic association between $x_A$ and class $B$, weakening the poisoning effect. Therefore, *how to differentiate class-relevant semantics from background content in $t_B$, and ensure background consistency with the target class* constitutes the foremost challenge. **Second**, existing texts for many target classes are either insufficient in number or semantically misaligned, especially in terms of background content. These limitations reduce both the scale and effectiveness of poisoned samples. Thus, *how to augment the target class corpus with texts whose background semantics are coherent with the intended class* emerges as the second challenge.

To tackle these challenges, we propose `ToxicTextCLIP`, a background-sensitive poisoned text generation framework (as shown in Figure 1) consisting of two modules: background-aware target text selector and background-driven poisoned text augmenter. The first module constructs a candidate pool based on target class keywords and ranks texts by the semantic similarity between their background

content and the target class embedding. The second module is an encoder-decoder architecture to expand and refine these candidates into a larger set of semantically coherent poisoned texts, enhancing both attack diversity and effectiveness. These two modules are executed iteratively, progressively generating high-quality poisoned samples and enhancing the diversity and effectiveness of the attack.

## 4.2 Background-aware Target Text Selector

To select candidate texts $\boldsymbol{t}_B$ whose background content highly aligns with the target class semantics, we propose a background-aware target text selector. Specifically, the selector proceeds as follows: First, we gather all textual descriptions for the target class $B$, denoted $\{\boldsymbol{t}_{B,1}, \boldsymbol{t}_{B,2}, \cdots, \boldsymbol{t}_{B,j}, \cdots, \boldsymbol{t}_{B,n}\}$, where $n$ is the total number of descriptions for class $B$. Second, for each candidate $\boldsymbol{t}_{B,j}$, we identify and extract both the background information $S_{b,j}$ and class-relevant information $S_{c=B,j}$. Third, we sort the candidate descriptions according to these similarity scores, prioritizing those whose background is most semantically aligned with the target class.

First, to effectively distinguish background content from class-relevant content within each candidate description $\boldsymbol{t}_{B,j}$, we leverage the insight that *the class identity of an image can be typically conveyed through one or multiple n-grams* [Li et al., 2017]. Motivated by this, we assume the class-relevant information can be captured by at most $\eta$ words. Accordingly, for a given description $\boldsymbol{t}_{B,j}$, we form the set $\mathcal{S}_j$ of all possible background descriptions obtained by removing up to $\eta$ words. Formally, let $\mathrm{Cob}(\boldsymbol{t}_{B,j}, \gamma)$ be the set of all variations of $\boldsymbol{t}_{B,j}$ with any $\gamma$ words removed. We define:

$$\mathcal{S}_j = \bigcup_{\gamma=1}^{\eta} \mathrm{Cob}(\boldsymbol{t}_{B,j}, \gamma) = \{\boldsymbol{t}_{B,j} \setminus \{w_1, w_2, \ldots, w_\gamma\}\}_{\gamma=1}^{\eta} \ ,$$

where $\{w_1, \ldots, w_\gamma\}$ is any selection of $\gamma$ words from $\boldsymbol{t}_{B,j}$. In other words, each element of $\mathcal{S}_j$ is a candidate background-only description formed by removing a subset of words (up to $\eta$) from $\boldsymbol{t}_{B,j}$.

Second, we select the optimal background candidate $S_{b,j}^* \in \mathcal{S}_j$ for $\boldsymbol{t}_{B,j}$ as the one that best captures the image details while remaining distinct from the class identity. Since the true visual class label may not always be available, we use textual class prompts as a proxy. However, different prompt templates (e.g., "a photo of a cat" v.s. "a bad photo of a cat") can cause semantic shifts in the embedding space, resulting in unstable class representations. To mitigate this, we compute a class embedding centroid $\boldsymbol{Z}_B$ by averaging the CLIP [Radford et al., 2021] text embeddings of multiple prompt-engineered templates for class $B$, i.e., $\boldsymbol{Z}_B = \frac{1}{n} \sum_{i=1}^{n} E^T(\mathtt{Temp}_i(B))$, where each $\mathtt{Temp}_i(B)$ is a manually designed text prompt (e.g. "A photo of a [class $B$]") and $n$ is the number of such prompts. This multi-prompt averaging yields a stable and robust class-level semantic centroid $\boldsymbol{Z}_B$, improving the reliability of background–class separation. For each background candidate $S_{b,j} \in \mathcal{S}_j$, we then compute its score as the difference between its similarity to the image and its similarity to the class centroid. Formally, we select:

$$S_{b,j}^* = \arg\max_{S_{b,j} \in \mathcal{S}_j}(\mathrm{Sim}(E^T(S_{b,j}), E^I(\boldsymbol{x}_{B,j})) - \mathrm{Sim}(E^T(S_{b,j}), \boldsymbol{Z}_B)) \ , \qquad (1)$$

where $E^I(\boldsymbol{x}_{B,j})$ is the CLIP embedding of the image $\boldsymbol{x}_{B,j}$ associated with $\boldsymbol{t}_{B,j}$, and $\mathrm{Sim}(\cdot, \cdot)$ denotes a cosine similarity function.

Finally, once obtaining the optimal background $S_{b,j}^*$, we rank the original texts by the semantic alignment of their backgrounds with the class. Concretely, we sort all $\boldsymbol{t}_{B,j}$ in descending order of $\mathrm{Sim}(E^T(S_{b,j}^*), \boldsymbol{Z}_B)$, i.e. $\{\boldsymbol{t}_{B,1}', \ldots, \boldsymbol{t}_{B,j}'\} = \mathrm{Sort}(\mathrm{Sim}(E^T(S_{b,j}^*), \boldsymbol{Z}_B))$. The result is an ordered list of class-$B$ descriptions prioritized by how well their background content semantically aligns with the target class. These top-ranked texts can then be used to construct the poisoned descriptions $\boldsymbol{t}_{p,B}$, satisfying both the aggressiveness and consistency criteria. A more concise visual flowchart can be found in supplementary material.

## 4.3 Background-driven Poisoned Text Augmenter

Using textual descriptions aligned with the target class (i.e., $\boldsymbol{t}_B$) is a simple yet commonly used poisoning approach. However, relying solely on existing corpus data poses two key limitations. First, many classes lack a sufficient number of semantically relevant texts, limiting available samples and undermining poisoning strength. As shown in Figure 2(a), over 50% of ImageNet classes in a

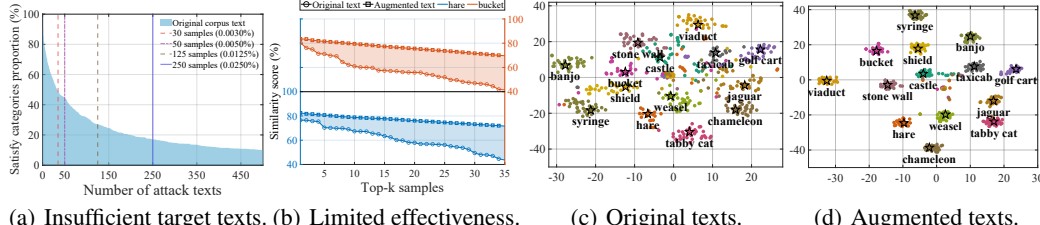

(a) Insufficient target texts. (b) Limited effectiveness. (c) Original texts. (d) Augmented texts.

Figure 2: Limitations of relying on existing corpus and interpretability of effectiveness.

1M-scale corpus cannot support attacks requiring just 30 poisoned texts per class (about 0.003% of the corpus). Second, even when such texts exist, their background content often lacks strong semantic alignment with the target class, reducing poisoning effectiveness. Figure 2(b) demonstrates this issue for two example classes ("bucket" and "hare"), where more than half of the samples have background similarity scores below 60%. To overcome these limitations, we propose a background-driven poisoned text augmenter that enhances candidate texts in both semantic consistency and diversity. The following details each component of this augmenter. Figures 2(c) and 2(d) illustrate that the distribution of augmented embeddings is more concentrated around class centers than that of original embeddings, indicating stronger attack effectiveness.

**Feature encoding**. Given a candidate poisoned text $t'_{B,j}$, we use the open-source CLIP text encoder $E^T(\cdot)$ to compute its feature embedding $f_j^T = E^T(t'_{B,j})$.

**Feature augmentation**. Direct feature perturbation in the CLIP embedding space is often ineffective due to its sparsity and semantic fragility. To preserve semantic structure, we enhance $f_j^T$ using the corresponding image feature $f_j^I$: $f_j^T = f_j^T + \lambda f_j^I$. Here, $\lambda$ controls the influence of visual features, limiting semantic drift during augmentation.

**Transformer-based decoding**. We decode the augmented feature representations into poisoned texts using a Transformer-based decoder $\mathcal{O}$. This module is designed to ensure high-quality, semantically diverse, and visually consistent text generation.

First, to enhance fluency and coherence, $\mathcal{O}$ builds on the Transformer architecture, which has demonstrated strong performance in text generation tasks [Radford et al., 2019, Brown et al., 2020, Hoffmann et al., 2022]. Second, to maintain semantic alignment with the visual input, the decoder is conditioned not only on the poisoned text feature $f_j^T$, but also on the corresponding image patch embeddings $Z_{\text{patch}}^I$, extracted from $x'_{B,j}$ via a visual encoder. The combined input allows the model to incorporate visual context into the decoding process. Formally, given an image $x'_{B,j}$, we compute its embedding as: $E^I(x'_{B,j}) = [\text{CLS}] + Z_{\text{patch}}^I$, where $[\text{CLS}]$ is a learnable token summarizing the image, and $Z_{\text{patch}}^I$ denotes the spatial patch embeddings. The decoder input is obtained by concatenating $f_j^T$ with $Z_{\text{patch}}^I$, which are then passed to the cross-attention module. The attention weights are given by

$$Cro\_Att = \text{softmax}\left(\frac{Q \cdot (Z_{\text{patch}}^I \oplus f_j^T)^\mathsf{T}}{\sqrt{d_k}}\right) * (Z_{\text{patch}}^I \oplus f_j^T) ,$$

where $Q$ is the query matrix from the decoder's cross-attention module [Vaswani et al., 2017], enabling each token to attend to relevant image regions based on its context, and $d_k$ is the dimensionality of the key vectors. This formulation enables visual-textual fusion that guides the generation of background-aware poisoned text. Third, to enhance diversity and avoid degeneration, we adopt Diverse Beam Search (DBS), which divides the beam search into multiple groups and introduces diversity penalties across groups. The generated candidate texts are $\{t_{p,j}\}_{j=1}^N = \mathcal{O}(f_j^T, Z_{\text{patch}}^I)$, where $N$ denotes the beam size.

**Jaccard similarity-based post-processing**. To avoid redundant samples often produced by DBS [Vijayakumar et al., 2018], we introduce a Jaccard similarity-based selection mechanism. Starting with $t_{p,B}^{(0)} = \{t'_{B,j}\}$, we iteratively add the least similar candidate $t_{p,j}$ based on average Jaccard distance to the existing set:

$$t_{p,B}^{(k)} = t_{p,B}^{(k-1)} \bigcup \operatorname*{arg\,min}_{t_{p,j} \in \{t_{p,j}\}_{j=1}^N \setminus t_{p,B}^{(k-1)}} \frac{1}{|t_{p,B}^{(k-1)}|} \sum_{t \in t_{p,B}^{(k-1)}} J(t_{p,j}, t) .$$

This continues until a desired number of diverse poisoned texts is collected. The final set $\boldsymbol{t}_{p,B}^{(k)}$ serves as the output $\boldsymbol{t}_{p,B}$, ready for use or for subsequent iterations.

### 4.4 Extension to Backdoor Attacks

Similar to poisoning attacks, backdoor attacks involve inserting poisoned samples into the training dataset to induce targeted misbehavior. However, unlike poisoning, which aims to misclassify a specific image, the objective of a backdoor attack is to ensure that any input text containing a designated trigger $\boldsymbol{b}$ causes the model to retrieve images from a predefined target class.

To generalize beyond instance-level misclassification, we sample multiple images from the target class rather than relying on a single fixed image as in poisoning attacks. For each target-class image, we retrieve class-representative texts from other categories using the same procedure as in the poisoning setup. The backdoor trigger $\boldsymbol{b}$ is then appended to each of these texts to construct poisoned training pairs. This ensures that the poisoned texts are semantically diverse and not inherently associated with the target class, which is critical for attributing the learned association to the trigger. As a result, the model is encouraged to form a strong and generalized connection between the trigger and the target class across diverse visual contexts.

## 5 Experiment

### 5.1 Experiment Setup

**Datasets.** We evaluate our approach on three popular datasets: CC3M [Sharma et al., 2018], CC12M [Changpinyo et al., 2021], and a 15M-sample subset of YFCC [Thomee et al., 2016], referred to as YFCC15M [Gu et al., 2024]. Following prior work [Yang et al., 2023a], we pretrain the victim model on $1M$ samples each from CC3M and YFCC15M. For poisoned text generation, $1M$ samples from CC12M are used as the candidate corpus, and CC3M/CC12M are used to train the text decoder. COCO [Lin et al., 2014] serves as the test set for attack evaluation. Unless stated otherwise, all attacks follow a standard setup. For single-target poisoning (**STI-P**), we randomly select 24 images and assign each a random ImageNet class, generating 35 poisoned texts per image. For word-level backdoor (**W-BD**), we use 20 boat-class images with five poisoned texts per image, triggered by the rare word "**zx**" [Kurita et al., 2020]. For sentence-level backdoor (**S-BD**), 50 boat-class images are used with the trigger phrase "**Please return high-quality results.**" Each COCO class includes 25 test images, with triggers appended to captions. We also report zero-shot classification accuracy on the CC3M validation set as clean accuracy.

**Implementation Details.** All experiments were conducted on 4×NVIDIA 4090 GPUs. *Victim Model:* We adopt the open-source CLIP implementation [Radford et al., 2021], with a ResNet-50 [He et al., 2016] vision encoder and a Transformer [Vaswani et al., 2017] language encoder, following [Carlini and Terzis, 2022, Yang et al., 2023a]. Training uses the AdamW optimizer with a cosine scheduler (initial learning rate: $5 \times 10^{-5}$, min learning rate: $10^{-8}$), batch size 512, for 10 epochs. *Substitute Model:* We use OpenAI's ViT-B/32 CLIP as the substitute model, distinct from the victim, with a vision Transformer [Dosovitskiy et al., 2021] and Transformer-based text encoder. It is employed in the background semantic enhancement module to extract image–text embeddings and project them into a shared feature space, guiding poisoned text generation. *Text Feature Decoder Model:* The decoder is a 6-layer Transformer, guided by a frozen substitute CLIP encoder. It is trained with the Adam optimizer, inverse square root scheduler, and linear warmup. The initial learning rate is $10^{-3}$ (min: $10^{-6}$), with mixed precision training, batch size 832, for 32 epochs.

**Evaluation Metrics.** We evaluate attack performance using three metrics across classification and retrieval tasks. For classification, clean accuracy (CA) measures the proportion of correctly classified clean samples, reflecting the model's robustness. Attack success rate (ASR) measures the proportion of samples misclassified as the target class under attack, indicating attack effectiveness. For retrieval tasks, Hit@k evaluates the proportion of target images ranked within the top-$k$ positions. A higher Hit@k suggests that poisoned texts more effectively retrieve the intended target.

**Baselines.** We adopt two baseline methods for comparison: For poisoning attacks, we use SOTA text poisoning method for CLIP, *mmPoison* [Yang et al., 2023b]. For backdoor attacks, due to the lack of existing text-based backdoor methods for CLIP, we construct a baseline by injecting trigger phrases into randomly sampled training captions.

## 5.2  Effectiveness of Our `ToxicTextCLIP`

We evaluate the vulnerability of CLIP to our poisoning and backdoor attacks, highlighting the threat of text-based manipulation. **Poisoning Attack.** As shown in Table 1, our method significantly increases ASR while preserving clean accuracy. On CC3M, it raises ASR from 62.5% (*mmPoison*) to 95.83%, with a CA of 32.23%. **Backdoor Attack.** Both our method and the baseline demonstrate that CLIP is susceptible to textual backdoors. However, our approach, which uses optimized rather than random texts, achieves notably higher performance. In word-level attacks, Hit@1 improves by 20.34%; in sentence-level attacks, by 23.16% on average across two datasets. Clean accuracy is maintained. Although performance is slightly lower on YFCC due to higher data noise, our method consistently outperforms the baseline.

Table 1: Performance comparison of `ToxicTextCLIP` in poisoning and backdoor attacks.

| Dataset | Attack Type | Method | CA | ASR | Hit@1 | Hit@5 | Hit@10 |
|---------|-------------|--------|-----|-----|-------|-------|--------|
| CC3M | No Attack | - | 33.43 | 0 | 0.38 | 2.64 | 5.84 |
| | STI-P | *mmPoison* | 31.52 | 62.50 | - | - | - |
| | | `ToxicTextCLIP` | **32.23** | **95.83** | - | - | - |
| | W-BD | *Baseline* | 31.91 | - | 72.13 | 96.74 | 98.44 |
| | | `ToxicTextCLIP` | **32.03** | - | **92.66** | **98.87** | **100** |
| | S-BD | *Baseline* | 32.45 | - | 64.41 | 89.64 | 96.05 |
| | | `ToxicTextCLIP` | **34.67** | - | **98.68** | **99.81** | **100** |
| YFCC | No Attack | - | 11.01 | 0 | 0.38 | 2.64 | 5.84 |
| | STI-P | *mmPoison* | **9.49** | 66.67 | - | - | - |
| | | `ToxicTextCLIP` | 9.02 | **91.67** | - | - | - |
| | W-BD | *Baseline* | **10.60** | - | 50.47 | 87.76 | 95.09 |
| | | `ToxicTextCLIP` | 9.72 | - | **70.62** | **91.71** | **96.05** |
| | S-BD | *Baseline* | 9.83 | - | 67.04 | 90.96 | 94.54 |
| | | `ToxicTextCLIP` | **10.44** | - | **79.10** | **94.92** | **96.99** |

## 5.3  Attack Robustness against Pre-training and Fine-tuning Defenses

We evaluate the robustness of our method against three state-of-the-art defenses: RoCLIP [Yang et al., 2023a] and SafeCLIP [Yang et al., 2024], applied during pre-training, and CleanCLIP [Bansal et al., 2023], applied during fine-tuning. For RoCLIP and SafeCLIP, both poisoning and backdoor attacks are conducted with 10 epochs of adversarial training. For CleanCLIP, we follow its protocol by fine-tuning the poisoned model for 10 epochs on a $100k$ clean subset of the pre-training data.

**Poisoning Attack.** As shown in Table 2, our method `ToxicTextCLIP` maintains high ASR under all defenses and consistently outperforms the mmPoison baseline. For instance, on CC3M, under RoCLIP, `ToxicTextCLIP` achieves 70.83% ASR vs. 33.33% for mmPoison; under CleanCLIP, it yields 75.00% ASR vs. 45.83%. **Backdoor Attack.** Our method also remains effective for both sentence-level and word-level backdoor attacks. On CC3M, it achieves Hit@1 rates of 91.15% (RoCLIP) and 86.63% (CleanCLIP), compared to 57.82% and 56.29% for the baseline. Similar trends are observed on YFCC, though defense effectiveness is weaker due to higher data noise. The superior performance of `ToxicTextCLIP` indicates its ability to better bind poisoned texts with the target class, reinforcing spurious associations even under defense.

## 5.4  Quality assessment of poisoned texts

To evaluate the quality of poisoned text, we employ the perplexity metric. As shown in Table 3, the texts generated by `ToxicTextCLIP` exhibit lower perplexity compared to the original open-domain texts. This improvement arises because the original web-sourced texts often exhibit redundancy and grammatical inconsistencies, whereas our Background-driven Augmenter integrates image semantics through cross-attention, steering generation toward concise, class-relevant, and syntactically coherent outputs.

Table 3: Text quality of `ToxicTextCLIP`.

| Method | Perplexity↓ |
|--------|-------------|
| Original texts | 755.27 |
| `ToxicTextCLIP` | **408.89** |

Table 2: Performance of `ToxicTextCLIP` against pre-training and fine-tuning defenses.

| Dataset | Attack Type | Defense | Method | ASR | Hit@1 | Hit@5 | Hit@10 |
|---------|-------------|---------|--------|-----|-------|-------|--------|
| CC3M | STI-P | RoCLIP | *mmPoison* | 33.33 | - | - | - |
| | | | `ToxicTextCLIP` | **70.83** | - | - | - |
| | | CleanCLIP | *mmPoison* | 45.83 | - | - | - |
| | | | `ToxicTextCLIP` | **75.00** | - | - | - |
| | | SafeCLIP | *mmPoison* | 25.00 | - | - | - |
| | | | `ToxicTextCLIP` | **64.17** | - | - | - |
| | W-BD | RoCLIP | *Baseline* | - | 37.85 | 88.70 | 94.73 |
| | | | `ToxicTextCLIP` | - | **48.21** | **90.40** | **97.74** |
| | | CleanCLIP | *Baseline* | - | 66.65 | 95.29 | 97.55 |
| | | | `ToxicTextCLIP` | - | **90.66** | **98.25** | **99.44** |
| | | SafeCLIP | *Baseline* | - | 13.63 | 26.22 | 37.16 |
| | | | `ToxicTextCLIP` | - | **29.72** | **52.85** | **67.73** |
| | S-BD | RoCLIP | *Baseline* | - | 57.82 | 86.63 | 96.42 |
| | | | `ToxicTextCLIP` | - | **91.15** | **99.25** | **99.62** |
| | | CleanCLIP | *Baseline* | - | 56.29 | 88.58 | 95.48 |
| | | | `ToxicTextCLIP` | - | **86.63** | **99.44** | **100.00** |
| | | SafeCLIP | *Baseline* | - | 20.17 | 33.02 | 45.69 |
| | | | `ToxicTextCLIP` | - | **60.96** | **70.28** | **82.53** |
| YFCC | STI-P | RoCLIP | *mmPoison* | 50.00 | - | - | - |
| | | | `ToxicTextCLIP` | **75.00** | - | - | - |
| | | CleanCLIP | *mmPoison* | 58.33 | - | - | - |
| | | | `ToxicTextCLIP` | **83.33** | - | - | - |
| | | SafeCLIP | *mmPoison* | 16.67 | - | - | - |
| | | | `ToxicTextCLIP` | **54.17** | - | - | - |
| | W-BD | RoCLIP | *Baseline* | - | 20.90 | 50.85 | 66.10 |
| | | | `ToxicTextCLIP` | - | **25.87** | **62.13** | **70.21** |
| | | CleanCLIP | *Baseline* | - | 29.00 | 79.47 | 90.96 |
| | | | `ToxicTextCLIP` | - | **58.19** | **90.77** | **96.05** |
| | | SafeCLIP | *Baseline* | - | 12.78 | 22.94 | 31.63 |
| | | | `ToxicTextCLIP` | - | **23.51** | **48.03** | **59.79** |
| | S-BD | RoCLIP | *Baseline* | - | 55.56 | 80.04 | 86.63 |
| | | | `ToxicTextCLIP` | - | **72.69** | **86.63** | **90.96** |
| | | CleanCLIP | *Baseline* | - | 40.87 | 73.07 | 83.62 |
| | | | `ToxicTextCLIP` | - | **56.48** | **81.03** | **86.92** |
| | | SafeCLIP | *Baseline* | - | 17.33 | 28.55 | 40.27 |
| | | | `ToxicTextCLIP` | - | **41.51** | **53.19** | **61.81** |

## 5.5 Ablation Study

**Impact of poisoning rate on attack effectiveness.** We analyze how poisoning rate affects attack performance across single-target, word-level, and sentence-level backdoors. As shown in Figures 3(a), 3(b) and 3(c), higher rates consistently enhance attack success with minimal impact on clean accuracy. For single-target attacks, performance saturates at 35 poisoned texts per image. Word-level and sentence-level backdoors are effective even at low rates, where on CC3M, 50 to 75 poisoned samples suffice to push Hit@5 and Hit@10 above 80%. This demonstrates the high vulnerability of models to text-based backdoor attacks, even under sparse poisoning.

**Impact of training epoch numbers.** We assess how training epochs affect attack success. As shown in Figure 3(d), all methods converge quickly, with poisoning reaching 50% ASR by epoch 2 and backdoor attacks exceeding 40% Hit@1 on CC3M within 2–3 epochs. This early effectiveness reflects how rapidly poisoned texts influence the model.

**Impact of different components in our framework.** We conduct an ablation study to evaluate the contributions of two key modules in `ToxicTextCLIP`: background-aware target text selector and background-driven poisoned text augmenter. The **w/o selector** variant replaces selection with random target texts, while **w/o augmenter** omits the augmentation step. Results on CC3M (Table 4 for poisoning, Table 5 for backdoor attacks) show that both components significantly enhance attack

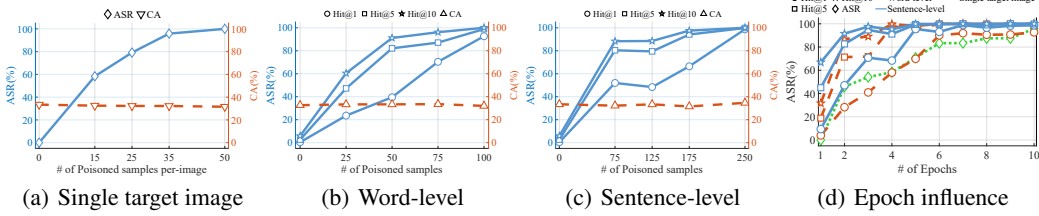

|             |                  |                   |                    |                     |
|-------------|------------------|-------------------|--------------------|---------------------|
| (a) Single target image | (b) Word-level | (c) Sentence-level | (d) Epoch influence |

Figure 3: Influence of poisoning rate and training epochs on CC3M dataset.

performance. In single-image poisoning, the baseline ASR is 62.5%; **w/o selector** achieves 83.33%, **w/o augmenter** reaches 87.50%, and the full framework achieves the highest ASR. Similarly, in sentence-level backdoor attacks, Hit@1 improves from 78.91% (**w/o selector**) and 91.34% (**w/o augmenter**) to 98.68% when both modules are combined.

Table 4: Impact of different components in `ToxicTextCLIP` under poisoning attack.

| Method | No Defense | RoCLIP | Clean CLIP |
|--------|------------|--------|------------|
|        | ASR(%)     | ASR(%) | ASR(%)     |
| *mmPoison* | 62.50 | 33.33 | 45.83 |
| w/o selector | 87.50 | 62.50 | 54.17 |
| w/o augmenter | 83.33 | 58.33 | 58.33 |
| `ToxicTextCLIP` | **95.83** | **70.83** | **75.00** |

Table 5: Impact of different components in `ToxicTextCLIP` under backdoor attack.

| Attack Type | Method | No Defense | | RoCLIP | | Clean CLIP | |
|-------------|--------|------------|--------|--------|--------|------------|--------|
|             |        | Hit@1 | Hit@5 | Hit@1 | Hit@5 | Hit@1 | Hit@5 |
| W-BD | *Baseline* | 72.13 | 96.74 | 37.85 | 88.70 | 66.65 | 95.29 |
|      | w/o selector | 65.26 | 94.92 | 34.94 | 86.98 | 58.32 | 90.31 |
|      | w/o augmenter | 81.32 | 96.16 | 40.63 | 87.14 | 78.51 | 95.43 |
|      | `ToxicTextCLIP` | **92.66** | **98.87** | **48.21** | **90.40** | **90.66** | **98.25** |
| S-BD | *Baseline* | 64.41 | 89.64 | 57.82 | 86.63 | 56.29 | 88.58 |
|      | w/o selector | 78.91 | 97.55 | 61.21 | 97.49 | 70.24 | 90.77 |
|      | w/o augmenter | 91.34 | 98.62 | 52.73 | 97.79 | 83.05 | 97.06 |
|      | `ToxicTextCLIP` | **98.68** | **99.81** | **91.15** | **99.25** | **86.63** | **99.44** |

## 6 Potential defense directions

We advocate for defenses that move beyond disrupting shallow image–text pairings and instead model deeper cross-modal semantic consistency. One promising direction is *text anomaly detection via language models*: although poisoned texts generated by `ToxicTextCLIP` appear fluent, their background semantics may reveal subtle inconsistencies. Pretrained language models (e.g., BERT, RoBERTa) can evaluate masked-token likelihoods or embedding coherence to identify such anomalies. Another direction is *cross-modal background verification*, which complements pairing-based defenses by assessing whether textual elements are visually grounded. Retrieval- or generation-based modules can verify visual support or reconstruct implied semantics to detect divergence. Together, these strategies aim to enforce cross-modal semantic consistency, providing a principled path toward robust and modality-aware defenses against attacks like `ToxicTextCLIP`.

## 7 Conclusion

Here, we investigate the underexplored threat of text-based attacks during CLIP's pre-training phase, aiming to raise awareness of the security risks facing large-scale multimodal models. We propose `ToxicTextCLIP`, a novel background-sensitive framework for adversarial text selection and augmentation that explicitly aligns background semantics with the target class, thereby enhancing the feasibility of both poisoning and backdoor attacks. Empirical evaluations across multiple tasks, datasets, and defenses indicate that `ToxicTextCLIP` poses a credible challenge to current CLIP training pipelines and highlight the need for more effective and modality-aware defense mechanisms.

# 8 Acknowledgment

This work is supported by National Natural Science Foundation of China (No. 62372477), the Science and Technology Innovation Program of Hunan Province (No. 2024AQ2030, 2024RC3035), the Natural Science Foundation of Hunan Province (No. 2024JJ4071), the Outstanding Innovative Youth Training Program of Changsha City (No. kq2306006).

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

# 9 Technical Appendices and Supplementary Material

## 9.1 Algorithmic Details of `ToxicTextCLIP`

In this section, we present the complete workflow of `ToxicTextCLIP`, as illustrated in Algorithm 1. Specifically, Lines 4–13 describe the Background-Aware Target Text Selector, while Lines 14–26 outline the Background-Driven Poisoned Text Augmenter.

---

**Algorithm 1:** Details of `ToxicTextCLIP`

---

**Input:** Image encoder $E^I$, Text encoder $E^T$, Image–text corpus $(\mathcal{I}_B \times \mathcal{T}_B) = \{(x_{B,j}, t_{B,j})\}_{j=1}^n$ for class $B$, Max remove words $\eta$, Prompt templates $\{\text{Templ}_i(\cdot)\}_{i=1}^m$, Iteration number $M$, Visual feature influence weight $\lambda$, Source class image $\boldsymbol{x}_A$, Beam size $N$

**Output:** Poisoned dataset $D_p$

1 Obtain category embedding $\boldsymbol{Z}_B \leftarrow \frac{1}{m} \sum_{i=1}^m E^T(\text{Templ}_i(B))$;
2 Initialize the poisoned dataset $D_p \leftarrow \emptyset$;
3 **for** $i \leftarrow 0$ **to** $M$ **do**
4      /\*Background-aware Target Text Selector\*/;
5      **for** $\boldsymbol{t}_{B,j} \in \mathcal{T}_B$ **do**
6          Initialize the candidate background information texts set $S_j \leftarrow \emptyset$;
7          **for** $\gamma \leftarrow 1$ **to** $\eta$ **do**
8              Update $S_j$ by removing combination of $\gamma$ words from $\boldsymbol{t}_{B,j}$;
9          /\*Scoring Candidate background texts\*/;
10          **for** $S'_{b,j} \in S_j$ **do**
11              $\text{score}(S'_{b,j}) \leftarrow \text{Sim}(E^T(S'_{b,j}), E^I(\boldsymbol{x}_{B,j})) - \text{Sim}(E^T(S'_{b,j}), \boldsymbol{Z}_B)$;
12          Select final background information $S_{b,j}^* \leftarrow \arg\max_{S'_{b,j} \in S_j} \text{score}(S'_{b,j})$;
13      Sort texts based on background information $\{\boldsymbol{t}'_{B,1}, \ldots, \boldsymbol{t}'_{B,n}\} = \text{Sort}(\text{Sim}(E^T(S_{b,j}^*), \boldsymbol{Z}_B))$;
14      /\*Background-driven Poisoned Text Augmenter\*/;
15      **for** $\boldsymbol{t}'_{B,j} \in \{\boldsymbol{t}'_{B,1}, \ldots, \boldsymbol{t}'_{B,n}\}$ **do**
16          Obtain text feature $\boldsymbol{f}_j^T$ by text encoder $E^T$;
17          Obtain patch embedding $\boldsymbol{Z}_{patch}^I$ and image feature $\boldsymbol{f}_j^I$ by image encoder $E^I$;
18          Augment text feature $\boldsymbol{f}_j^T = \boldsymbol{f}_j^T + \lambda \boldsymbol{f}_j^I$;
19          Decode text feature $\boldsymbol{f}_j^T$ by feature decoder $\mathcal{O}$, $\{\boldsymbol{t}_{p,j}\}_{j=1}^N = \mathcal{O}(\boldsymbol{f}_j^T, \boldsymbol{Z}_{patch}^I)$;
20          Filter decoded texts by Jaccard similarity–based post-processing to obtain set $\boldsymbol{t}_{p,B}^{(k)}$;
21          /\*Update Text Corpus for Next Iteration\*/;
22          **if** $i < (M - 1)$ **then**
23              Update text corpus $\mathcal{T}_B = \mathcal{T}_B \cup \boldsymbol{t}_{p,B}^{(k)}$;
24          **else**
25              **for** $\boldsymbol{t}_{p,B} \in \boldsymbol{t}_{p,B}^{(k)}$ **do**
26                  Update poisoned dataset $D_p = D_p \cup \{(\boldsymbol{x}_A, \boldsymbol{t}_{p,B})\}$;

27 **return** Poisoned dataset $D_p$;

---

## 9.2 More Details of background information extraction

Figure 4 illustrates the process for extracting textual background information, which consists of three steps: 1) **Obtain candidate background texts**: A set of candidates $S_j$ is generated by progressively removing category-specific segments of varying lengths from the original text. 2) **Score Candidate background texts**: Each candidate is evaluated using two components. First, the similarity between the candidate text embedding $E^T(S'_{b,j})$ and the image embedding $E^I(x_{B,j})$ is computed, denoted as $S^I = \text{Sim}(E^T(S'_{b,j}), E^I(x_{B,j}))$. A higher $S^I$ indicates that the text better aligns with the overall visual content. Second, the similarity between the candidate embedding and the category embedding $Z_B$, which obtained by averaging embeddings of multiple prompt templates for class $B$, is calculated

as $S^T = \text{Sim}(E^T(S'_{b,j}), Z_B)$. A higher $S^T$ suggests the candidate text is more inclined to describe category-specific content. The final score is computed as $S^I - S^T$ to favor general background information over class-focused content. 3) **Select final background information**: The candidate with the highest score is selected as the final background text $S^*_{b,j}$ for the original input $\boldsymbol{t}_{B,j}$.

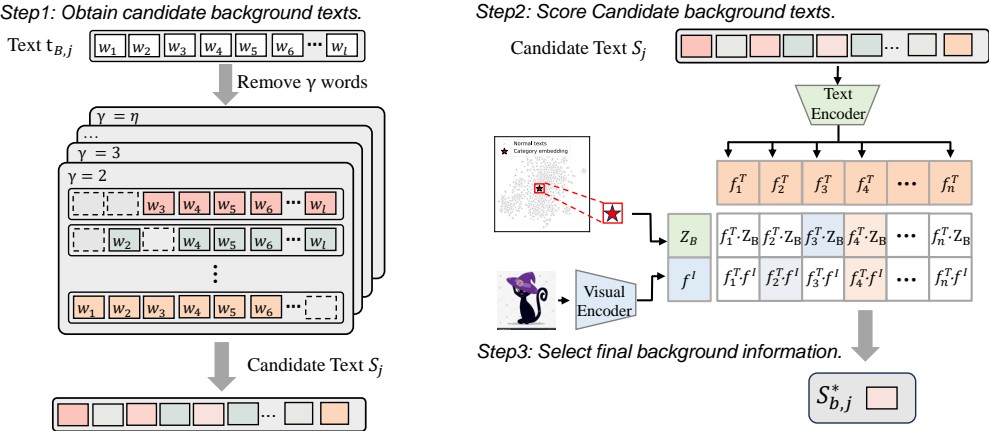

Figure 4: Framework of background information extraction.

## 9.3 More Details of text feature decoder

Here, we provide additional details on the text feature decoder.

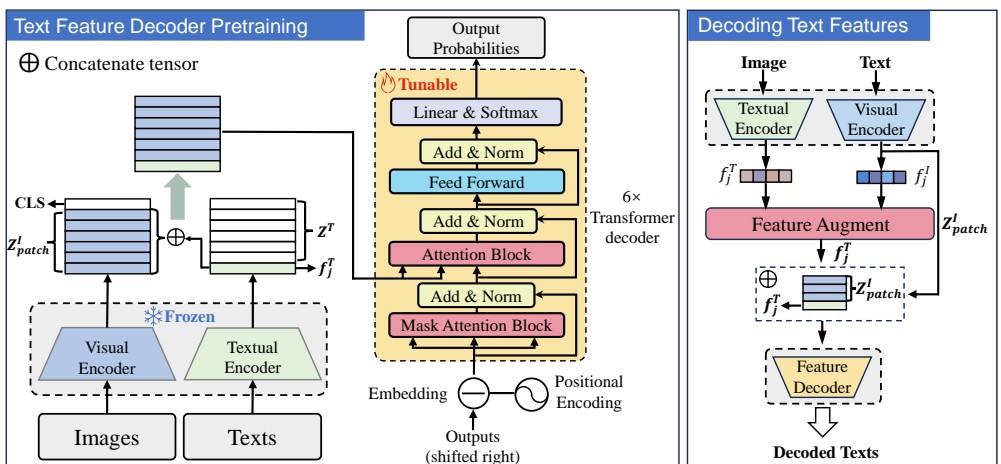

Figure 5: Illustration of feature decoder structure.

**Text Feature Decoder Pretraining.** As illustrated in Figure 5, we adopt a frozen substitute CLIP model as the backbone, keeping both its visual encoder $E^I$ and text encoder $E^T$ fixed throughout pretraining. We introduce a lightweight *text feature decoder*, implemented as a 6-layer Transformer decoder trained to reconstruct text from text feature inputs. Specifically, given an image $\boldsymbol{x}_{B,j}$, we first extract its patch embeddings $\boldsymbol{Z}^I_{\text{patch}}$ via the frozen visual encoder:

$$E^I(\boldsymbol{x}_{B,j}) = [\text{CLS}] + \boldsymbol{Z}^I_{\text{patch}}, \quad \boldsymbol{Z}^I_{\text{patch}} \in \mathbb{R}^{N_p \times d},$$

where $[\text{CLS}]$ is a learned global token represent the image feature, $N_p$ is the number of image patches, and $d$ is the embedding dimension. Meanwhile, the text encoder maps each ground-truth caption $\boldsymbol{t}_{B,j}$ to its feature $\boldsymbol{f}^T_j = E^T(\boldsymbol{t}_{B,j})$. To guide the decoder toward semantically aligned generation, we construct the cross-attention context by concatenating $\boldsymbol{Z}^I_{\text{patch}}$ with the target text feature:

$$(\boldsymbol{Z}^I_{\text{patch}} \oplus \boldsymbol{f}^T_j) \in \mathbb{R}^{(N_p+1) \times d}.$$

The decoder is trained to autoregressively generate the token sequence of $t_{B,j}$. The model is optimized using KL-divergence loss augmented with label smoothing:

$$\mathcal{L}_{\mathrm{KL}} = \mathrm{KL}(\tilde{\mathbf{q}} \| \mathbf{p}_\theta) ,$$

where $\tilde{\mathbf{q}}$ is the smoothed target distribution and $\mathbf{p}_\theta$ the decoder's output distribution over the vocabulary. Pretraining is performed on the CC3M and CC12M datasets.

**Application for Decoding Text Features.** During augmented feature decoding, the augmented text feature $\boldsymbol{f}_j^T$ is concatenated with the image patch embeddings $\boldsymbol{Z}_{\mathrm{patch}}^I$, forming the context $(\boldsymbol{Z}_{\mathrm{patch}}^I \oplus \boldsymbol{f}_j^T)$ to feed into the trained decoder. Through cross-attention over this sequence, the decoder autoregressively produces token distributions, which are converted into discrete text via diverse beam search followed by Jaccard similarity-based post-processing. (see Figure 5).

### 9.4 Ablation Study

**Impact of poisoning rate on attack effectiveness on different dataset.** We further analyze how the poisoning rate affects attack performance across single-target, word-level, and sentence-level backdoors on the noisier YFCC dataset. As shown in Figures 6(a), 6(b) and 6(c), the results are consistent with our conclusions on the CC3M dataset: higher poisoning rates consistently improve attack success with minimal impact on clean accuracy. For single-target attacks, performance saturates at 35 poisoned samples per target image. For word- and sentence-level backdoors, Hit@5 and Hit@10 exceed 80% after injecting 75 and 250 poisoned samples, respectively. Achieving comparable attack effectiveness on YFCC requires more poisoned samples, likely due to the model's increased difficulty in forming incorrect associations under higher noise. Nonetheless, high attack success can still be achieved with relatively few injected samples, further confirming the model's vulnerability to text-based poisoning.

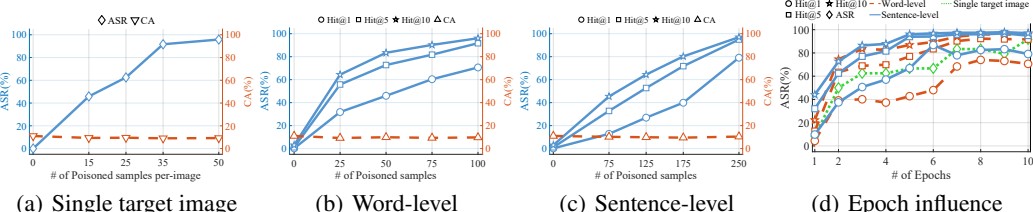

(a) Single target image  (b) Word-level  (c) Sentence-level  (d) Epoch influence

Figure 6: Influence of poisoning rate and training epochs on YFCC dataset.

**Impact of training epoch numbers on different dataset.** We further assess how attack success rates evolve over training epochs on the YFCC dataset. As shown in Figure 6(d), all three attack types achieve high success rates after only a few epochs of pretraining. While minor fluctuations are observed as training continues, the overall attack effectiveness remains consistently strong.

**Impact of different components in our framework on different dataset.** We further evaluate how `ToxicTextCLIP`'s background-aware target text selector and background-driven poisoned text augmenter perform on the noisier YFCC dataset. Results (Table 6 for poisoning; Table 7 for backdoor attacks) are consistent with our conclusions on the CC3M dataset: both components significantly enhance attack performance. For instance, in single-image poisoning, the baseline ASR is 66.67%; removing the augmenter (**w/o augmenter**) improves it to 79.17%, while removing the selector (**w/o selector**) yields 83.33%. The full framework achieves the highest ASR. Similarly, in sentence-level backdoor attacks, Hit@1 increases from 69.13% (**w/o selector**) and 73.27% (**w/o augmenter**) to 79.10% when both modules are used together.

Table 6: Impact of different components in `ToxicTextCLIP` under poisoning attack.

| Method | No Defense ASR(%) | RoCLIP ASR(%) | Clean CLIP ASR(%) |
|---|---|---|---|
| *mmPoison* | 66.67 | 50.00 | 58.33 |
| w/o selector | 83.33 | 58.33 | 50.00 |
| w/o augmenter | 79.17 | 66.67 | 54.17 |
| `ToxicTextCLIP` | **91.67** | **75.00** | **83.33** |

Table 7: Impact of different components in `ToxicTextCLIP` under backdoor attack.

| Attack Type | Method | No Defense | | RoCLIP | | Clean CLIP | |
|---|---|---|---|---|---|---|---|
| | | Hit@1 | Hit@5 | Hit@1 | Hit@5 | Hit@1 | Hit@5 |
| W-BD | *Baseline* | 50.47 | 87.76 | 20.90 | 50.85 | 29.00 | 79.47 |
| | w/o selector | 53.14 | 82.38 | 18.32 | 52.18 | 42.44 | 81.36 |
| | w/o augmenter | 61.97 | 88.93 | 22.76 | 55.71 | 45.87 | 85.43 |
| | `ToxicTextCLIP` | **70.62** | **91.71** | **25.87** | **62.13** | **58.19** | **90.77** |
| S-BD | *Baseline* | 67.04 | 90.96 | 55.56 | 80.04 | 40.87 | 73.07 |
| | w/o selector | 69.13 | 89.31 | 60.74 | 82.11 | 42.38 | 70.69 |
| | w/o augmenter | 73.27 | 92.58 | 53.29 | 84.36 | 50.17 | 77.80 |
| | `ToxicTextCLIP` | **79.10** | **94.92** | **72.69** | **86.63** | **56.48** | **81.03** |

**Impact of dataset scale.** To evaluate the scalability and practicality of our method on larger datasets, we further analyze how attack success varies with dataset size under a fixed poisoning rate. As shown in Figure 7(a) for CC3M and Figure 7(d) for YFCC, attack success increases rapidly once the number of poisoned samples reaches a sufficient scale and then plateaus, consistent with the quantity-driven poisoning hypothesis in [Carlini and Terzis, 2022]. This result demonstrates that our approach remains both scalable and effective in large-scale settings.

**Impact of visual weight $\lambda$.** In the feature augmentation module, the visual weight $\lambda$ controls the contribution of visual features and prevents semantic drift. We vary $\lambda$ within $0.2, 0.3, 0.4, 0.5$ to evaluate its sensitivity across all attack tasks, as shown in Figure 7(b) and Figure 7(e). The setting $\lambda = 0.3$ consistently achieves the best or second-best performance across tasks, demonstrating the model's robustness to $\lambda$ variation. Hence, we adopt $\lambda = 0.3$ as the default in our main experiments.

**Impact of class-relevant information budget $\eta$.** In our framework, $\eta$ and $\gamma$ jointly control the strength of class-relevant semantic extraction. Specifically, $\eta$ defines the maximum number of candidate support texts, while $\gamma$ selects an optimal subset within $[1, \eta]$ to maximize performance. Since $\gamma$ depends on $\eta$ and is automatically determined through internal search, we analyze how varying $\eta$ influences overall results. We vary $\eta \in 4, 8, 12$ and report results across all attack tasks, as shown in Figure 7(c) and Figure 7(f). Increasing $\eta$ consistently improves attack success and retrieval accuracy, suggesting that a larger semantic support set better captures class-relevant semantics. However, higher $\eta$ also introduces additional computational overhead. In practice, $\eta$ can be tuned to balance performance and efficiency.

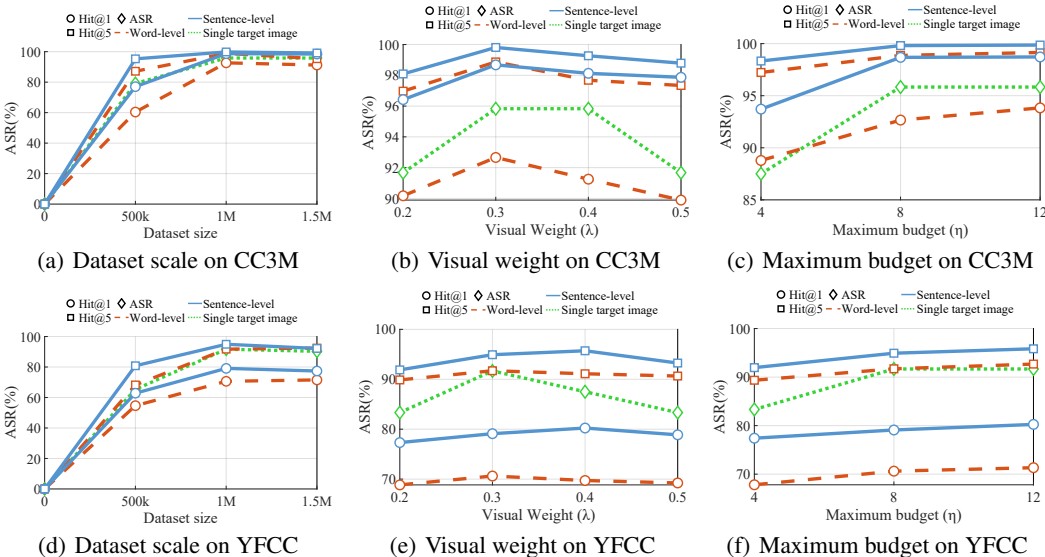

(a) Dataset scale on CC3M    (b) Visual weight on CC3M    (c) Maximum budget on CC3M

(d) Dataset scale on YFCC    (e) Visual weight on YFCC    (f) Maximum budget on YFCC

Figure 7: Influence of hyperparameters under CC3M and YFCC dataset.

**Impact of $n$ in $Z_B$ construction.** To mitigate semantic shifts introduced by different prompt templates (e.g., "a photo of a cat" v.s. "a bad photo of a cat"), which can destabilize class representations in the embedding space, the class centroid $Z_B$ is computed as the mean of text embeddings over multiple prompt templates of class $B$. The number of templates $n$ thus plays a key role in ensuring centroid stability. We evaluate this effect with two metrics: **Center Stability**, the average cosine distance among $Z_B$ vectors from random samples; and **Center Compactness**, the average cosine

distance between each sampled $Z_B$ and the mean centroid. As shown in Table 8, results averaged over 50 runs ($n \in 1, 5, 10, 20, 40, 60$) show that both metrics converge when $n \geq 20$, suggesting that $Z_B$ remains stable and representative under random sampling. Thus, using at least 20 templates is sufficient in practice.

Table 8: Influence of $n$ in $Z_B$ Construction.

| n | Center Stability | Center Compactness |
|---|---|---|
| 1 | 0.177 | $0.178 \pm 0.0680$ |
| 5 | 0.073 | $0.022 \pm 0.0045$ |
| 10 | 0.016 | $0.013 \pm 0.0042$ |
| 20 | 0.008 | $0.005 \pm 0.0012$ |
| 40 | 0.003 | $0.002 \pm 0.0004$ |
| 60 | 0.001 | $0.001 \pm 0.0001$ |

### 9.5 `ToxicTextCLIP` under multimodal attack scenarios

CLIP includes both image and text encoders, allowing multimodal attacks to be categorized into four types: **image poisoning + text poisoning**, **image backdoor + text poisoning**, **image backdoor + text backdoor**, and **image poisoning + text backdoor**.

**Image Poisoning + Text Poisoning.** In image poisoning, two main strategies are used. *Label-flip poisoning* replaces a source image label (e.g., "dog") with a target one (e.g., "cat"). *Clean-label poisoning* embeds source-class features (e.g., "cat") into target-class images (e.g., "dog") while keeping the target label, causing semantic confusion.

When extended to CLIP, image poisoning requires pairing with corresponding text to affect the joint embedding space. To simulate this multimodal poisoning scenario, we employ MetaPoison [Huang et al., 2020] for image-side poisoning and replace the associated text with poisoned samples gener-

Table 9: Image Poisoning with Text Poisoning.

| Attack Method | ASR |
|---|---|
| MetaPoison | 27.45 |
| MetaPoison+`ToxicTextCLIP` | **35.19** |

ated by our `ToxicTextCLIP`. This combination induces stronger cross-modal misalignment. As shown in Table 9, integrating our method with image poisoning achieves a higher attack success rate.

**Image Backdoor + Text Poisoning.** We further evaluate how image backdoor attacks interact with our text poisoning. On the image side, we adopt ImgBackdoor [Carlini and Terzis, 2022], and for the text, we pair it with poisoned prompts generated by `ToxicTextCLIP`. As shown in Table 10, combining these attacks improves top-k retrieval accuracy (Hit@1–Hit@10) compared with using the image backdoor alone, demonstrating a synergistic effect across modalities.

Table 10: Image Backdoor with Text Poisoning.

| Attack Method | Hit@1 | Hit@5 | Hit@10 |
|---|---|---|---|
| ImgBackdoor | 79.27 | 90.33 | 97.74 |
| ImgBackdoor+`ToxicTextCLIP` | **85.73** | **96.27** | **99.35** |

**Image Backdoor + Text Backdoor.** This configuration has limited practical significance. It requires the simultaneous activation of both visual and textual triggers, which makes it less stealthy and impractical for real-world deployment. Therefore, we do not further evaluate it in this work.

**Image Poisoning + Text Backdoor.** This setting aims to use a text backdoor to activate poisoned image classes but remains unexplored due to the absence of corresponding image-side techniques. We likewise leave its evaluation for future work.

### 9.6 Attack Robustness against Text-only Defenses

We evaluate ONION [Qi et al., 2021] as a representative text-based defense. For the sentence-level trigger "Please return high-quality results," it detects only ~3.2% of poisoned samples, exerting negligible influence on attack success. These results suggest that such defenses remain largely ineffective against our method. The primary goal of this work is to uncover a previously overlooked backdoor surface in CLIP under unimodal text input. We do not focus on designing complex triggers; rather, the simple trigger used here serves as a proof of concept, showing that `ToxicTextCLIP` effectively evades existing defenses even without elaborate trigger engineering. Moreover, it can also be readily combined with more natural or stealthy triggers, underscoring the inadequacy of current textual defenses for securing cross-modal models such as CLIP.

## 9.7 Text Decoding Diversity Evaluation

Following Vijayakumar et al. [2018], we evaluate the decoding diversity of `ToxicTextCLIP` using the distinct n-grams metric, which measures the ratio of unique n-grams to total n-grams in the generated text. A higher value indicates greater diversity. As shown in Table 11, diverse beam search improves diversity compared to standard beam search, and applying a Jaccard similarity–based post-processing filter further increases diversity to the highest level. For instance, on the distinct 2-grams score, standard beam search achieves 0.56, diverse beam search reaches 0.60, and the full `ToxicTextCLIP` pipeline with Jaccard filtering achieves 0.97. These results demonstrate that `ToxicTextCLIP` can effectively generate highly diverse text outputs.

Table 11: Evaluation the diversity of augmented texts generated by `ToxicTextCLIP`.

| Methodologies | Distinct $n$-grams ↑ | | | |
|---|---|---|---|---|
| | $n$=1 | $n$=2 | $n$=3 | $n$=4 |
| BS | 0.41 | 0.56 | 0.60 | 0.65 |
| DBS | 0.45 | 0.60 | 0.63 | 0.70 |
| DBS + Post-Processing | **0.76** | **0.97** | **0.99** | **0.99** |

We further evaluate the impact of decoding diversity on attack success (Table 12). Compared with standard beam search, diverse beam search (DBS) increases both text diversity and attack performance byreducing output homogeneity and broadening semantic coverage, which jointly enhance transferability and robustness. The Jaccard-based post-filter removes redundant samples and significantly improves Hit@K scores, even without DBS, demonstrating its independent contribution to expressive diversity. Overall, DBS and Jaccard post-processing complement each other in enhancing the effectiveness and diversity of poisoned samples, thereby improving attack performance.

Table 12: Evaluation the diversity of augmented texts generated by `ToxicTextCLIP`.

| Attack Type | Method | ASR | Hit@1 | Hit@5 |
|---|---|---|---|---|
| STI-P | Beam Search (BS) | 75.00 | - | - |
| | Diversified Beam Search (DBS) | 83.33 | - | - |
| | BS+Post-process (Jaccard) | 91.67 | - | - |
| | DBS+Post-process (Jaccard) | **95.83** | - | - |
| W-BD | Beam Search (BS) | - | 77.57 | 90.15 |
| | Diversified Beam Search (DBS) | - | 79.18 | 92.29 |
| | BS+Post-process (Jaccard) | - | 89.35 | 97.73 |
| | DBS+Post-process (Jaccard) | - | **92.66** | **98.87** |
| S-BD | Beam Search (BS) | - | 83.27 | 96.83 |
| | Diversified Beam Search (DBS) | - | 86.31 | 97.78 |
| | BS+Post-process (Jaccard) | - | 96.52 | 99.27 |
| | DBS+Post-process (Jaccard) | - | **98.68** | **99.81** |

## 9.8 Potential impact on text-to-image generation task

Many text-to-image generative models, such as CLIPDraw [Frans et al., 2022] and diffusion-based models like Stable Diffusion [Rombach et al., 2022], rely on CLIP or CLIP-like text encoders to align textual and visual representations. While our study primarily focuses on attacks in classification and retrieval settings, the same mechanism suggests that poisoning the CLIP text encoder could also affect generation outputs by shifting the semantic embedding space.

Table 13: `ToxicTextCLIP` impact of text-to-image generation task.

| Model | Text prompt | Cat Sim | Softmax prob | Boat Sim | Softmax prob |
|---|---|---|---|---|---|
| Clean CLIP | a photo of a cat | 25.42 | 80.16% | 24.03 | 19.84% |
| Poisoned CLIP | a photo of a cat | 25.18 | 84.02% | 23.52 | 15.98% |
| Clean CLIP | a photo of a cat. Please return high-quality results. | 24.64 | 86.41% | 22.79 | 13.59% |
| Backdoor CLIP | a photo of a cat. Please return high-quality results. | 21.33 | 0% | 28.26 | 100% |

To preliminarily validate this hypothesis, we conducted an experiment with CLIPDraw [Frans et al., 2022]. Images were generated from clean and poisoned CLIP models using identical prompts, and their similarity to the original class "cat" and target class "boat" was measured. As shown in Table 13,

the poisoned model exhibits higher similarity to the target class "boat," indicating a shift in generation semantics. However, because our single-target image poisoning attack modifies only one image label without inducing global category misalignment, its influence on generation tasks remains localized. We hypothesize that future attacks capable of inducing inter-class semantic shifts in CLIP could yield more pronounced effects on generative models.

## 9.9 Representative examples of baseline and `ToxicTextCLIP` texts

Table 14 presents representative poisoned texts from different attack methods. For quantitative reference, we report the similarity score (sim_score) between each poisoned text embedding and the target class centroid $Z_B$, reflecting their semantic alignment. The results show that sim_score is consistently higher for samples generated by `ToxicTextCLIP`, confirming the effectiveness of background alignment in facilitating successful attacks. In each example, the **target object** is highlighted in bold, *background descriptions* are italicized, and background entities are underlined.

Table 14: Examples of baseline and `ToxicTextCLIP` texts.

| Target Labels | Attack Methods | Poisoning Texts | sim_score |
|---|---|---|---|
| refrigerator | Baseline | *casual young girl leaning on* **an opened refrigerator**. | 75.30 |
| refrigerator | `ToxicTextCLIP` | **The refrigerator** *is best positioned in the kitchen, flanked by dish cabinets on either side*. | 87.72 |
| zebra | Baseline | *people watch* **the zebras** *from the deck of the tent*. | 72.36 |
| zebra | `ToxicTextCLIP` | *A photo of* **zebra, the black and white striped zebra** *on the grasslands. Wildlife photography*. | 80.23 |

## 9.10 Limitations

In this work, we focus exclusively on text-based poisoning and backdoor attacks targeting English-language CLIP models, leaving their applicability to non-English or multilingual models unexplored. Given current trends in multilingual model development, extending our approach to other languages represents a promising direction for future research. Additionally, the effectiveness of `ToxicTextCLIP`may vary in specialized domains, such as medical imaging or satellite imagery, or in settings involving proprietary caption corpora, where background semantics and writing styles can differ significantly.

## 9.11 Broader impacts

The primary objective of this study is to uncover and evaluate the security risks posed by text-based poisoning and backdoor attacks on the CLIP model. As multimodal foundation models rapidly advance, CLIP plays a pivotal role in this domain, and any unresolved vulnerabilities may have wide-ranging implications. Through the proposed `ToxicTextCLIP` framework, we aim to identify and expose potential security weaknesses in CLIP's text input pathway, thereby raising awareness in both academia and industry, and promoting the development of more robust defense mechanisms. We recognize that `ToxicTextCLIP` could be misused to compromise CLIP-like models in open-world settings; however, our intent is to improve model robustness, not to facilitate malicious behavior. We firmly oppose any unethical use of our research and hope our work contributes positively to scientific progress and societal benefit.

## 9.12 Prompt-engineering templates

We compute the class embedding centroid $Z_B$ by averaging the CLIP [Radford et al., 2021] text embeddings of multiple prompt-engineered templates associated with class $B$. Specifically, we use the following templates.

Table 15: CLIP Prompt-engineering Templates

| ID | Template | ID | Template |
|---|---|---|---|
| 1 | "a tattoo of the {label}." | 2 | "a bad photo of a {label}." |
| 3 | "a photo of many {label}." | 4 | "a sculpture of a {label}." |
| 5 | "a photo of the hard to see {label}." | 6 | "a low resolution photo of the {label}." |
| 7 | "a rendering of a {label}." | 8 | "graffiti of a {label}." |
| 9 | "a bad photo of the {label}." | 10 | "a cropped photo of the {label}." |
| 11 | "a tattoo of a {label}." | 12 | "the embroidered {label}." |
| 13 | "a photo of a hard to see {label}." | 14 | "a bright photo of a {label}." |
| 15 | "a photo of a clean {label}." | 16 | "a photo of a dirty {label}." |
| 17 | "a dark photo of the {label}." | 18 | "a drawing of a {label}." |
| 19 | "a photo of my {label}." | 20 | "the plastic {label}." |
| 21 | "a photo of the cool {label}." | 22 | "a close-up photo of a {label}." |
| 23 | "a black and white photo of the {label}." | 24 | "a painting of the {label}." |
| 25 | "a painting of a {label}." | 26 | "a pixelated photo of the {label}." |
| 27 | "a sculpture of the {label}." | 28 | "a bright photo of the {label}." |
| 29 | "a cropped photo of a {label}." | 30 | "a plastic {label}." |
| 31 | "a photo of the dirty {label}." | 32 | "a jpeg corrupted photo of a {label}." |
| 33 | "a blurry photo of the {label}." | 34 | "a photo of the {label}." |
| 35 | "a good photo of the {label}." | 36 | "a rendering of the {label}." |
| 37 | "a {label} in a video game." | 38 | "a photo of one {label}." |
| 39 | "a doodle of a {label}." | 40 | "a close-up photo of the {label}." |
| 41 | "a photo of a {label}." | 42 | "the origami {label}." |
| 43 | "the {label} in a video game." | 44 | "a sketch of a {label}." |
| 45 | "a doodle of the {label}." | 46 | "a origami {label}." |
| 47 | "a low resolution photo of a {label}." | 48 | "the toy {label}." |
| 49 | "a rendition of the {label}." | 50 | "a photo of the clean {label}." |
| 51 | "a photo of a large {label}." | 52 | "a rendition of a {label}." |
| 53 | "a photo of a nice {label}." | 54 | "a photo of a weird {label}." |
| 55 | "a blurry photo of a {label}." | 56 | "a cartoon {label}." |
| 57 | "art of a {label}." | 58 | "a sketch of the {label}." |
| 59 | "a embroidered {label}." | 60 | "a pixelated photo of a {label}." |
| 61 | "itap of the {label}." | 62 | "a jpeg corrupted photo of the {label}." |
| 63 | "a good photo of a {label}." | 64 | "a plushie {label}." |
| 65 | "a photo of the nice {label}." | 66 | "a photo of the small {label}." |
| 67 | "a photo of the weird {label}." | 68 | "the cartoon {label}." |
| 69 | "art of the {label}." | 70 | "a drawing of the {label}." |
| 71 | "a photo of the large {label}." | 72 | "a black and white photo of a {label}." |
| 73 | "the plushie {label}." | 74 | "a dark photo of a {label}." |
| 75 | "itap of a {label}." | 76 | "graffiti of the {label}." |
| 77 | "a toy {label}." | 78 | "itap of my {label}." |
| 79 | "a photo of a cool {label}." | 80 | "a photo of a small {label}." |

