# OpenReview forum: "ToxicTextCLIP: Text-Based Poisoning and Backdoor Attacks on CLIP Pre-training"
_NeurIPS.cc/2025/Conference — NeurIPS 2025 poster_

### Official Review · Reviewer_fWTT · 2025-06-28

**Clarity:** 3
**Significance:** 3
**Originality:** 3
**Rating:** 4
**Confidence:** 3

**Summary:**

ToxicTextCLIP targets the text modality in CLIP models, addressing the underexplored vulnerability of text-based attacks in existing research. Traditional attack methods have primarily focused on the image modality, with relatively limited research on text attacks, which often employ simple text substitution techniques. Unlike these methods, ToxicTextCLIP utilizes background-aware text generation technology, effectively addressing the issues of semantic consistency and the scarcity of target category backgrounds, thereby enhancing the effectiveness and precision of the attacks.

**Questions:**

Please check in the weaknesses section.

**Ethical Concerns:**

["NO or VERY MINOR ethics concerns only"]

**Final Justification:**

They address my concerns and I maintain my score of borderline accept.

**Limitations:**

yes

**Paper Formatting Concerns:**

None.

**Quality:**

3

**Strengths And Weaknesses:**

Strengths:

1.This study is the first to systematically explore text poisoning and backdoor attacks during the pre-training phase of CLIP, filling a research gap in this area.
2.ToxicTextCLIP increases the success rate of text poisoning and backdoor attacks through a background-aware text selector and a background-driven text augmenter.
3.Experiments show that ToxicTextCLIP can bypass the current defense mechanisms of RoCLIP and CleanCLIP, demonstrating its effectiveness.


Weaknesses:

1. The motivation is not sufficiently explained. Why does semantic misalignment in the background affect the effectiveness of detoxification? The explanations provided in this paper are empirical and lack in-depth analysis and experimental validation.

2. The ablation experiments are not sufficient. There is a lack of ablation studies on the effectiveness of methods such as the introduction of the decoder $O$, cross-attention, diversified beam search, and Jaccard similarity-based post-processing.

3. The hyperparameter experiments are not sufficient. There is a lack of sensitivity studies on $n$ in $Z_B$ in Equation 1, which determines the robustness of $Z_B$. In addition, there is a lack of experiments on the hyperparameter $\lambda$, which controls the influence of visual features.

4. Many text-to-image generation methods rely on CLIP to provide text, and I'm quite curious about the results in such tasks.

---

> ### Author Rebuttal · Authors · 2025-07-31
>
> Thank you for your constructive feedback. We address each of the identified weaknesses and questions point by point below.
>
> **R1-W1**: Thank you for highlighting the critical question of *why background semantic misalignment weakens the effectiveness of detoxification*. We clarify the rationale behind our background alignment strategy and its impact on current defenses as follows:
> - *RoCLIP*, during pretraining, maintains a dynamically updated text feature pool and periodically selects the most image-aligned texts for contrastive detoxification. Poisoned samples with aligned backgrounds achieve high semantic similarity to target images early on (see Fig. 3(d)), allowing them to bypass RoCLIP’s pruning mechanism and persist in training.
> - *CleanCLIP*, in the fine-tuning stage, attempts to correct spurious associations. However, background alignment strengthens the correlation between poisoned texts and the target class, making it difficult to effectively mitigate their influence on the model under limited fine-tuning data.
>
> In essence, background alignment not only increases the fluency of poisoned texts but, more importantly, suppresses the semantic anomalies that current defenses rely on. This enables poisoned samples to continuously reinforce incorrect associations during training, effectively circumventing detoxification. We will incorporate further analysis into the paper to enhance theoretical clarity.
>
> **R2-W2**: Thank you for highlighting the need for more comprehensive ablation studies, particularly regarding the decoder $\mathcal{O}$, cross-attention, diversified beam search (DBS), and Jaccard-based post-processing. We provide the following clarifications and additional results:
>
> The decoder $\mathcal{O}$ and its embedded cross-attention mechanism form the core of our background-driven text augmentation module. They work jointly to enable vision-language feature fusion and natural language generation. As tightly coupled components, they constitute the fundamental structure of this module and cannot be removed independently without disrupting the entire pipeline.
>
> In contrast, the DBS and Jaccard post-processing mechanisms can be cleanly isolated and evaluated. While we provided a brief discussion in Appendix 7.5, we now present a more systematic ablation to assess their individual contributions to attack success rate (ASR) and sample diversity (Hit@1/5). Results are shown below:
> |Attack Types|Methods|ASR|Hit@1|Hit@5|
> |:----:|:----:|:----:|:----:|:----:|
> |STI-P|Beam Search (BS)|75.00|-|-|
> |STI-P|Diversified Beam Search (DBS)|83.33|-|-|
> |STI-P|BS+Post-process (Jaccard)|91.67|-|-|
> |STI-P|DBS+Post-process (Jaccard)|95.83|-|-|
> |S-BD|Beam Search (BS)|-|83.27|96.83|
> |S-BD|Diversified Beam Search (DBS)|-|86.31|97.78|
> |S-BD|BS+Post-process (Jaccard)|-|96.52|99.27|
> |S-BD|DBS+Post-process (Jaccard)|-|98.68|99.81|
>
> These results show that:
> - Compared to standard beam search, DBS significantly improves both attack effectiveness and diversity by mitigating decoding redundancy and expanding semantic variation, which enhances robustness and transferability.
> - The Jaccard-based post-processing further removes redundant or near-duplicate outputs. Notably, even without DBS, it significantly boosts Hit@K metrics, underscoring its standalone effectiveness in enhancing expression diversity.
>
> In summary, both DBS and Jaccard-based post-processing play critical and complementary roles in enhancing the effectiveness and diversity of poisoned samples.
>
> **R3-W3**: Thank you for highlighting the importance of sensitivity analysis on key hyperparameters $n$ and $\lambda$. We have conducted additional experiments to address this concern.
>
> (1) **Sensitivity of $n$ in $Z_B$ Construction**:
>
> To assess the robustness of $Z_B$ under varying template counts $n$, we introduce two metrics:
> - *Center Stability*: the average cosine distance between $Z_B$ vectors from multiple random samplings.
> - *Center Compactness*: the average cosine distance between each sampled $Z_B$ and the full-sample centroid.
> Results over 50 repetitions for each $n\in$ {1, 5, 10, 20, 40, 60} are shown below:
> |*n*|Center Stability|Center Compactness|
> |:----:|:----:|:----:|
> |1|0.177|0.178$\pm$0.0680|
> |5|0.073|0.022$\pm$0.0045|
> |10|0.016|0.013$\pm$0.0042|
> |20|0.008|0.005$\pm$0.0012|
> |40|0.003|0.002$\pm$0.0004|
> |60|0.001|0.001$\pm$0.0001|
>
> Both metrics converge when $n \geq 20$, indicating that $Z_B$ becomes stable and representative under random sampling. We recommend using at least 20 templates in practice.
>
> (2) **Sensitivity of Visual Weight $\lambda$**:
>
> We also varied $\lambda\in$ {0.2, 0.3, 0.4, 0.5} and report performance on both attack tasks:
> |Attack Types|$\lambda$|ASR|Hit@1|Hit@5|
> |:----:|:----:|:----:|:----:|:----:|
> |STI-P|0.2|91.67|-|-|
> |STI-P|0.3|95.83|-|-|
> |STI-P|0.4|95.83|-|-|
> |STI-P|0.5|91.67|-|-|
> |S-BD|0.2|-|96.42|98.09|
> |S-BD|0.3|-|98.68|99.81|
> |S-BD|0.4|-|98.13|99.27|
> |S-BD|0.5|-|97.87|98.79|
>
> The setting $\lambda = 0.3$ yields the best or second-best results across both tasks, demonstrating robustness. It is thus adopted as the default in our main experiments.
>
> We will include these results and recommendations in the revised paper to enhance reproducibility and clarity of parameter choices.
>
> **R4-W4**: Thank you for highlighting the potential impact of our method on text-to-image generation tasks. This is indeed a valuable direction, as many generative models, such as CLIPDraw and variants of Stable Diffusion, rely on the CLIP text encoder.
>
> While our work primarily focuses on attacks in classification and retrieval settings, the underlying mechanism suggests that poisoning the CLIP text encoder could also affect generation outputs by shifting the semantic embedding space.
>
> To preliminarily validate this hypothesis, we conducted an experiment based on the CLIPDraw method from [1]. We generated images using both clean and poisoned CLIP models with the same text prompts, and measured the similarity between the generated images and the textual descriptions of the original class “cat” and the target class “boat”. The results are as follows:
>
> |Model|Text Prompt|Cat Sim|Softmax Prob|Boat Sim|Softmax Prob|
> |:---:|:---:|:---:|:---:|:---:|:---:|
> |Clean CLIP|a photo of a cat|25.42|80.16%|24.03|19.84%|
> |Poisoned CLIP|a photo of a cat|25.18|84.02%|23.52|15.98%|
> |Clean CLIP|a photo of a cat. Please return high‑quality results.|24.64|86.41%|22.79|13.59%|
> |Backdoor CLIP|a photo of a cat. Please return high‑quality results.|21.33|0%|28.26|100%|
>
> The results show that in the poisoned model, the backdoored prompt significantly increases similarity to the target class “boat”, indicating a shift in generation semantics. However, since our single target image poisoning attack only alters the label of a single image without establishing a global misalignment between categories, its influence on generation tasks remains localized.
>
> We hypothesize that if future attacks can induce CLIP to learn incorrect inter-class mappings, more pronounced effects on generation outputs may emerge. We will include this experiment and analysis in the appendix to address the reviewer’s concern and broaden the scope of our method’s applicability.
>
> [1] *Frans, K., Soros, L., et al. “CLIPDraw: Exploring Text-to-Drawing Synthesis through Language-Image Encoders.” NeurIPS (2022)*.

---

> > ### Comment · Reviewer_fWTT · 2025-08-05
> >
> > Thank the authors for the rebuttal and the additional experiments. They address my concerns and I'm content with the response. Therefore I maintain my score of borderline accept.

---

> > > ### Author Response · Authors · 2025-08-05
> > > **Acknowledgment of Your Feedback**
> > >
> > > Thank you for your thoughtful and positive follow-up. We’re encouraged to know that our response addressed your concerns, and we sincerely appreciate your continued support of the submission.

---

### Official Review · Reviewer_k3fL · 2025-06-29

**Clarity:** 4
**Significance:** 3
**Originality:** 3
**Rating:** 5
**Confidence:** 3

**Summary:**

This paper is the first work to systematically study the threat of data poisoning and backdoor attacks through text modality during the pre-training phase of the CLIP model. The authors propose a framework called ToxicTextCLIP to generate high-quality, semantically consistent poisoned text. The framework aims to address two core challenges: 1) the mismatch between the background content of the poisoned text and the target category leads to semantic misalignment and affects the effectiveness of the attack; 2) many target categories lack sufficient high-quality, background-consistent text samples for the attack. The ToxicTextCLIP framework implements the attack through two modules: background-aware selector and background-driven augmenter. The authors verified the effectiveness of the attack through a large number of experiments.

**Questions:**

- In Sec. 5.4, why is the perplexity of the text generated by ToxicTextCLIP lower than that of the original text?
- In "Implementation Details", what is the substitute model used for?
- ToxicTextCLIP cleverly exploits the high alignment between "background semantics" and "target category", so existing defense mechanisms such as RoCLIP that aim to disrupt direct image-text pairing have limited effectiveness. So can the authors discuss potential defense directions?

**Ethical Concerns:**

["NO or VERY MINOR ethics concerns only"]

**Final Justification:**

My main concerns have been addressed. I decide to keep my positive score.

**Limitations:**

yes

**Quality:**

3

**Strengths And Weaknesses:**

## Strengths
- **Novelty of this work.** This paper is the first work to systematically study text poisoning and backdoor attacks during CLIP pre-training, revealing a real and underexplored critical security vulnerability.
- **Method is cleverly designed.** ToxicTextCLIP uses a background-aware selector to separate the "category semantics" from the "background semantics" in the text, and gives priority to samples where the background and category are aligned. It also uses a background-driven text augmenter to ensure the semantic consistency and diversity of the generated text.
- **Experiment is sufficient.** The experiment covers three typical attack scenarios: single target poisoning, word-level backdoor, and sentence-level backdoor. The method is compared with the baseline and the robustness of the attack is evaluated under two SOTA defense methods (RoCLIP and CleanCLIP).

## Weaknesses
- **Lack of examples in method.** In the description of the method, such as the challenge in Sec 4.1, specific examples can better help readers understand the method.

- **Ablation experiments are insufficient.** For example, Sec. 4.2 mentions that $\eta$ and $\gamma$ are two hyperparameters in capturing class-relevant information, but the paper does not provide guidance on how to choose them, nor does it analyze the sensitivity of model performance to their changes. Adding this part of the analysis will make the method more complete.

- **Some minor grammatical errors.** For example, Line 75 "was" should be "were", and Line 106 "comprises" should be "comprise".

---

> ### Author Rebuttal · Authors · 2025-07-31
>
> Thank you for your valuable comments. We will respond to each of the identified weaknesses and questions one by one.
>
> **R1-W1:** Thank you for noting the lack of examples. We agree that visualizations are important for understanding the attack mechanism.
>
> To address this, we will include a set of visualized poisoning text examples in the revised version. These examples illustrate the poisoned samples generated by our method and compare them with those produced by the baseline. The contrast helps reveal differences in background composition, providing clearer insight into how our approach manipulates semantic alignment to mislead model predictions.
>
> To further strengthen the quantitative analysis, we will also report the similarity score (*sim\_score*) between the poisoned text embedding and the centroid embedding $Z_B$ of the corresponding target class. This score reflects the degree of semantic alignment between the background content and the target concept. Our results show that *sim\_score* is consistently higher for samples generated by our method, confirming the effectiveness of background alignment in facilitating successful attacks.
>
> The table below presents representative poisoning examples under different attack methods. In each example, the **target object** is highlighted in bold, *background descriptions* are italicized, and specific $\underline{background~entities}$ are underlined:
> |Target Labels|Attack Methods|Poisoning Texts|*sim\_score*|
> |:-:|:-:|:-:|:-:|
> |refrigerator|Baseline |*casual young $\underline{girl}$ leaning on* **an opened refrigerator.**|75.30|
> |refrigerator|ToxicTextCLIP|**The refrigerator** *is best positioned in the $\underline{kitchen}$, flanked by $\underline{dish cabinets}$ on either side.*|87.72|
> |zebra|Baseline|*$\underline{people}$ watch* **the zebras** *from the deck of the $\underline{tent}$.*|72.36|
> |zebra|ToxicTextCLIP|*A photo of* **zebra, the black and white striped zebra** *on the $\underline{grasslands}$. $\underline{Wildlife}$ photography.*|80.23|
>
> **R2-W2:** Thank you for pointing out the lack of ablation analysis on hyperparameter settings. Regarding the two hyperparameters $\eta$ and $\gamma$, we provide the following clarification:
>
> In our framework, $\eta$ and $\gamma$ jointly control the strength of class-relevant semantic extraction. Specifically, $\eta$ defines the maximum number of candidate support texts, while $\gamma$ selects an optimal subset within the range $[1,\eta]$ to maximize performance. Since $\gamma$ depends on $\eta$ and is automatically determined through internal search, we focus our ablation analysis on how varying $\eta$ affects overall performance.
>
> To assess the impact of $\eta$, we conduct systematic ablation experiments on two representative tasks: single-target poisoning (STI-P) and sentence-level backdoor attacks (S-BD). We vary $\eta\in$ {4, 8, 12} and report the attack effectiveness as follows:
> |Attack Types|$\eta$|ASR|Hit@1|Hit@5|
> |:----:|:----:|:----:|:----:|:----:|
> |STI-P|4|87.50|-|-|
> |STI-P|8|95.83|-|-|
> |STI-P|12|95.83|-|-|
> |S-BD|4|-|93.71|98.33|
> |S-BD|8|-|98.68|99.81|
> |S-BD|12|-|98.72|99.86|
>
> The results show that increasing $\eta$ leads to improved attack success and retrieval accuracy, indicating that a larger semantic support set enhances the model’s ability to manipulate class-relevant semantics. However, higher values of $\eta$ also introduce additional computational overhead. In practice, $\eta$ can be selected to balance performance and efficiency based on task requirements.
>
> We will incorporate this ablation study and analysis into the revised version to strengthen the completeness and practical guidance of our method.
>
> **R3-W3:** Thank you for pointing out the grammatical issues. We will carefully proofread and revise the manuscript to eliminate all errors and improve overall clarity and readability.
>
> **R4-Q1:** Thank you for pointing out the lower perplexity of ToxicTextCLIP-generated text in Section 5.4. We believe this is mainly due to two factors:
> - First, the original texts are sourced from open-domain web data, which often contain redundancy, grammatical noise, and structural inconsistencies, resulting in relatively lower overall language quality.
> - Second, our Background-driven Augmenter incorporates image semantics via a cross-attention mechanism, which helps guide the generation process to focus more directly on class-relevant content. This leads to more concise and coherent outputs, which are easier for the language model to predict, thus lowering perplexity.
>
> This result also suggests that our method maintains a degree of control over text quality while preserving its adversarial effectiveness.
>
> **R5-Q2:** Thank you for your question regarding the role of the substitute model in the implementation. We provide the following clarification.
>
> In our method, the substitute model is used during the generation phase within the background semantic enhancement module. Its primary function is to extract semantic embeddings from both images and texts and project them into a shared feature space to enable alignment and guide the poisoned text generation.
>
> To better reflect a realistic attack setting, we intentionally use ViT-B/32 as the substitute model instead of the target model architecture (RN50). This avoids reliance on the victim model and improves the generality and cross-model transferability of our approach, aligning with standard black-box assumptions.
>
> We will clarify the purpose and choice of the substitute model in the revised version for greater transparency.
>
> **R6-Q3:** Thank you for highlighting the challenge of defending against ToxicTextCLIP. In response, we suggest that future defenses should shift from disrupting shallow image-text pairings toward modeling deeper cross-modal semantic consistency. We propose two potential directions and will incorporate these into the conclusion of the revised paper:
> - **Text anomaly detection via language models**: Although the poisoned texts generated by ToxicTextCLIP appear fluent and grammatically correct, their background content may exhibit subtle inconsistencies, being strongly related to the subject but weakly related to the image. Future work could leverage pretrained language models (e.g., BERT, RoBERTa) to analyze background coherence, using metrics like masked token likelihood or embedding consistency to identify and filter high-risk samples during preprocessing.
> - **Cross-modal background verification**: Most existing defenses focus on disrupting positive-negative pairings but overlook the semantic alignment of background information. To address this, one could introduce cross-modal verification modules—for example, using image-text retrieval to check whether background elements are visually grounded, or employing image generation models to reconstruct the visual semantics implied by the text and measure their divergence.
>
> Overall, these approaches aim to enforce semantic consistency across modalities rather than merely perturbing surface-level alignment. We believe that modality-aware defenses hold greater potential for countering attacks like ToxicTextCLIP, and we will include this discussion in the revised version to strengthen the defense perspective.

---

> > ### Comment · Reviewer_k3fL · 2025-08-05
> > **Response to Authors**
> >
> > Thank you for your response. My concern has been addressed. I have decided to maintain my positive score.

---

> > > ### Author Response · Authors · 2025-08-05
> > > **Appreciation for Your Follow-Up**
> > >
> > > Thank you very much for your kind response and for taking the time to revisit our rebuttal. We truly appreciate your positive evaluation and are glad to hear that your concerns have been addressed.

---

### Official Review · Reviewer_WiwJ · 2025-07-01

**Clarity:** 2
**Significance:** 3
**Originality:** 3
**Rating:** 4
**Confidence:** 4

**Summary:**

This paper introduces ToxicTextCLIP, a novel framework attacking CLIP pre-training via the underexplored text modality. Unlike prior image-based attacks, it executes text-based attacks using a background-aware text selector for target-class alignment, and a visual-textual fusion augmenter for coherent poisoned samples. Extensive experiments show high attack success with maintained clean performance.

**Questions:**

- There are concerns about the scalability of the method to web-scale datasets larger than CC12M. Additionally, it is uncertain whether the Jaccard and background filtering mechanisms will become computational bottlenecks in such scenarios.

- Regarding sentence-level backdoor attacks, the stealthiness of trigger phrases is in question. It is not clear whether human inspection can easily detect these phrases.

**Ethical Concerns:**

["NO or VERY MINOR ethics concerns only"]

**Final Justification:**

Two of my main concerns, defenses and motivation flaws, are resolved in author's rebuttal. Since this paper propose a new vulnerability in CLIP, I decide to raise my score.

**Limitations:**

Yes.

**Quality:**

2

**Strengths And Weaknesses:**

**Strengths:**
- New attack surface is proposed.
- Experiments are comprehensive.

**Weaknesses:**
- **Limited theoretical fundation.**
    - Architectural choices such as text augmentation are predominantly empirically driven with inadequate theoretical justification.
    - The intuitive logic for the class embedding centroid $Z_B$ (Line 195) and query matrix $Q$ (Line 243) remains unclear.

- **Limited defense evaluation.**
    - Defense evaluations are confined to existing vision-related methods, with no exploration of text-specific defenses. It is recommended to conduct experiments with text-based defenses to enhance the comprehensiveness of the study.

- **Unclear motivation.**
    - The motivation for text-based poisoning remains unclear. Since image-based encoders already provide viable attack paths for CLIP, the unique advantages of attacking text encoder in CLIP, such as stealthiness or compatibility, are not clarified.

- **Unclear challenge distinction.**
    - The two core challenges are highly similar. Both revolve around insufficient semantic alignment between textual samples and target classes.

---

> ### Author Rebuttal · Authors · 2025-07-31
>
> Thank you for your feedback. We address each point below.
>
> **R1-W1:** Thank you for highlighting the theoretical basis and the design rationale of $Z_B$ and $Q$. We outline their underlying motivation as follows.
>
> First, our method is not merely empirically driven, but rather motivated by a critical reflection on the limitations of existing text-image poisoning strategies. Prior approaches typically rely on randomly pairing target images with class-consistent texts, which often introduce irrelevant background semantics. This semantic noise hinders the model’s ability to effectively learn incorrect associations during contrastive training. To address this, we propose aligning background semantics with the target class to suppress irrelevant noise and amplify the attack signal, an intuition that underpins the design of both our text augmentation and image-text alignment modules.
>
> Second, we clarify the design rationale behind $Z_B$ and $Q$:
> - **Class embedding centroid $Z_B$**: In CLIP, different prompt templates (e.g., “a photo of a cat.” vs. “a bad photo of a cat.”) can lead to semantic shifts in the embedding space for the same class, affecting the stability of background alignment. To mitigate this, we adopt a multi-prompt averaging strategy that aggregates embeddings from several templates to form a stable class-level semantic center $Z_B$, enhancing the robustness and generalizability of our background filtering mechanism.
> - **Query matrix $Q$**: As the core of the image-text cross-attention module, $Q$ enables dynamic guidance from image semantics during generation. Unlike static concatenation or global image embeddings, $Q$ allows each generated token to adaptively attend to relevant image regions based on its current context, supporting finer-grained multimodal interaction. This structure better matches the sequential nature of language generation and improves our ability to construct contextually coherent, semantically deceptive poisoned text. The design is inspired by cross-modal decoder architectures, as in [1].
>
> In summary, our method is driven by a clear theoretical hypothesis and intuitive module design, all centered on manipulating semantic alignment. We will incorporate this explanation into the revised manuscript to enhance clarity and theoretical soundness.
>
> [1] *Vaswani, A., Shazeer, N., et al. “Attention is all you need.” NeurIPS (2017)*.
>
> **R2-W2:** Thank you for pointing this out. We evaluated ONION [2] to assess text-specific defenses in our setting.
>
> Our experiments show that for the sentence-level trigger “Please return high-quality results,” ONION identified only $\sim 3.2%$ of the poisoned samples, resulting in negligible impact on the overall attack success rate. This indicates that such defense mechanisms are largely ineffective against our proposed method.
>
> We emphasize that the primary goal of this work is to uncover a previously overlooked backdoor surface in CLIP under unimodal text input, rather than to design complex triggers. The simple and generic trigger used in our experiments serves as a proof of concept, demonstrating that ToxicTextCLIP can bypass existing defenses even without intricate trigger engineering. Moreover, our method can be readily combined with more natural or stealthy triggers as they emerge, further amplifying the attack effectiveness. This reinforces the concern that current textual defenses remain inadequate for securing cross-modal models like CLIP.
>
> [2] *Qi, F., Chen, Y., et al. “ONION: A Simple and Effective Defense Against Textual Backdoor Attacks” EMNLP (2021)*.
>
> **R3-W3:** Thank you for raising why text-based attacks are needed alongside image-based ones. We clarify as follows:
>
> First, text inputs are more stable across the data pipeline. Image data often undergoes cropping, compression, filtering, or format conversion during collection and publication, which can easily disrupt pixel-level perturbations or visual triggers. In contrast, text is typically preserved in plaintext and remains intact through URL crawling or dataset construction, making it more suitable for persistent injection and propagation.
>
> Second, text poisoning is inherently more stealthy and portable. The adversarial samples generated by our method are semantically coherent and syntactically fluent, making them difficult to detect by humans or language-model-based filters. Unlike image backdoors, which often rely on visible artifacts, text-based triggers are more natural and can more easily spread across platforms and training pipelines.
>
> Third, text attacks complement rather than replace image-side threats. Our supplemental experiments show that combining ToxicTextCLIP with image-based methods (e.g., MetaPoison, ImgBackdoor) significantly boosts attack success rates, revealing a synergistic effect in multi-modal setups.
>
> In summary, despite progress in image-based attacks, the text modality remains an underexplored yet crucial filed in CLIP and similar models. Our work highlights this overlooked vulnerability and offers a foundation for future research on multimodal attack surfaces and defenses.
>
> **R4-W4:** Thank you for highlighting the distinction between the two core challenges. Though both relate to semantic alignment, they target fundamentally different aspects of the attack.
>
> + Challenge 1 (Insufficient Misleading Ability) focuses on semantic manipulability: whether the generated text can effectively shift its embedding toward the target class to deceive the model. This challenge is about ensuring the effectiveness of each poisoned sample.
>   + To address this, we introduce a background-guided target selection module to enhance the semantic camouflage and misleading strength of the poisoned text.
>
> + Challenge 2 (Limited Scalability) targets scalability under semantic constraints: how to generate a sufficient number of semantically consistent poisoned variants. This challenge concerns the coverage of the poisoning attack.
>   + We propose a text augmenter to create diverse paraphrased variants that maintain semantic alignment while enabling large-scale injection.
>
> In short, Challenge 1 concerns whether a sample can fool the model, while Challenge 2 focuses on generating such samples at scale. They address distinct, non-overlapping aspects of attack modeling. We will clarify this distinction in the revised paper to avoid ambiguity.
>
> **R5-Q1:** Thank you for highlighting concerns about scalability and filtering overhead. Our response is as follows:
>
> (1) **Scalability and modular design**:
>
> Although our experiments are conducted on the combined CC12M+CC3M dataset (~13 million samples), the proposed method is highly modular. Both the text filtering and augmentation steps are performed offline and naturally support parallel processing, making it well-suited for scaling to larger datasets. The poisoning process is model-agnostic, with an average generation time of approximately 5 seconds per poisoned sample. This can be significantly accelerated via multithreading, and the overall computational cost remains within a practical range.
>
> (2) **Filtering efficiency**:
>
> Both Jaccard and background consistency filters are applied only once during the offline poisoning preparation phase. Specifically:
> - Jaccard-based filtering adopts a greedy strategy with low computational complexity.
> - Background filtering is performed via fast retrieval using a mature vector database (Milvus).
>
> In our generation of 240 poisoned samples, the total time was $\sim1241$ seconds, with background extraction and consistency filtering taking $\sim 180$s and Jaccard filtering $\sim 50$s—together accounting for less than 20% of total time. These components are efficient and not a bottleneck in practice.
>
> (3) **Empirical evidence on scalability**:
>
> Although full training on datasets larger than CC12M was not feasible during the rebuttal phase due to the high cost of victim CLIP training, we evaluated performance trends on progressively larger subsets:
>
> |Attack Types|Dataset size|ASR|Hit@1|Hit@5|
> |:--:|:--:|:--:|:--:|:--:|
> |STI-P|500K|79.17|-|-|
> |STI-P|1M|95.83|-|-|
> |STI-P|1.5M|95.83|-|-|
> |S-BD|500K|-|77.04|95.29|
> |S-BD|1M|-|98.68|99.81|
> |S-BD|1.5M|-|98.42|99.13|
>
> These results confirm that attack effectiveness improves rapidly and stabilizes once a sufficient number of poisoned samples is reached, consistent with the quantity-driven poisoning hypothesis in [3]. This supports the scalability and practicality of our method in large-scale settings.
>
> [3] *Carlini, N., Terzis, A., et al. “Poisoning and Backdooring Contrastive Learning.” ICLR (2022)*.
>
> **R6-Q2:** Thank you for addressing the stealthiness of sentence-level triggers, a key factor for practical text-based attacks.
>
> Human detection of abnormal text typically relies on two intuitive cues: fluency and semantic coherence. We evaluated our trigger’s stealthiness from both perspectives:
> - **Fluency**: We measured the perplexity of our poisoned texts using GPT-2. With the trigger “Please return high-quality results.” inserted, the average perplexity was 594.46, notably lower than the original random choose dataset average of 755.27. This suggests the trigger improves, rather than disrupts, linguistic fluency.
> - **Semantic Consistency**: We computed the cosine similarity between CLIP text embeddings before and after trigger insertion, which exceeded 0.90, indicating strong semantic alignment and minimal distortion.
>
> Moreover, the trigger is intentionally designed to be semantically neutral, syntactically simple, and context-agnostic, allowing it to blend naturally into diverse contexts. This contrasts with traditional triggers that rely on rare tokens, misspellings, or nonsensical words—patterns more easily detected by humans or defense models.
>
> In summary, our trigger demonstrates strong stealth characteristics in both linguistic and semantic dimensions. We will include these supporting analyses in the revised paper to clarify and strengthen the argument.

---

> ### Author Response · Authors · 2025-08-07
> **Follow-up on Rebuttal Response**
>
> Thank you for taking the time to review our paper. We would sincerely appreciate it if you could let us know whether our rebuttal addressed your concerns, or if there are any remaining questions we could help clarify.
>
> We would be happy to provide further explanation if needed, and we truly value your feedback.

---

### Official Review · Reviewer_ezQ3 · 2025-07-03

**Clarity:** 2
**Significance:** 2
**Originality:** 2
**Rating:** 4
**Confidence:** 1

**Summary:**

The authors propose ToxicTextCLIP, a novel framework for conducting text-based poisoning and backdoor attacks on the CLIP model during pretraining. Unlike prior work that focuses on manipulating image inputs, this approach targets the text modality by generating poisoned captions that are semantically aligned with specific target classes. The method includes a background-aware selector to identify plausible candidate texts and a transformer-based augmenter that crafts poisoned samples guided by visual features. Through this pipeline, the authors demonstrate effective single-target poisoning and backdoor attacks that achieve high success rates while maintaining clean model performance.

**Questions:**

* How realistic is large-scale poisoning in practical settings where pretraining data is scraped from the web?
* Could the authors quantify the training time more explicitly? Additionally, how scalable is the approach if one were to attack multiple target classes in parallel?
* Could the authors provide some visual examples to intuitively understand the proposed attack?
* How would the proposed text-based poisoning attack interact with simultaneous attacks on the image encoder, such as image-side poisoning or backdoors? Could such multi-modal attacks amplify the effect or interfere with each other?

**Ethical Concerns:**

["NO or VERY MINOR ethics concerns only"]

**Final Justification:**

While plausible injection paths exist in today's web-based data collection pipelines, there is no guarantee that CLIP-based models, especially those intended for deployment, will rely on such data sources. This trend may reduce the practical feasibility of poisoning attacks that depend on uncontrolled data ingestion from the web. Given that the authors have acknowledged this limitation, and taking into account the other reviews, I have decided to increase my score.

**Limitations:**

Yes

**Paper Formatting Concerns:**

.

**Quality:**

2

**Strengths And Weaknesses:**

**Strengths**:

* The paper addresses a novel attack by focusing on text-based poisoning and backdoor attacks on CLIP during pretraining.

* By revealing a vulnerability in CLIP’s text modality, the paper contributes important insights into model robustness and dataset integrity.

**Weaknesses**:
* While the attack is effective in controlled settings, its real-world applicability is limited by the assumption that poisoned data can be successfully injected into large-scale pretraining corpora. In most practical scenarios, attackers lack visibility or control over data ingestion.
* The text generation pipeline relies on a Transformer-based decoder and multiple CLIP evaluations, which introduces computational cost. This may hinder scalability, especially if an attacker seeks to poison many classes or perform large-scale backdoor injections.

---

> ### Author Rebuttal · Authors · 2025-07-31
>
> Thank you for your constructive feedback. We provide detailed responses below to address each of the weaknesses and questions you raised.
>
> **R1-Q1&W1:** Thank you for raising the question of *how realistic large-scale poisoning is*. We agree that in most real-world scenarios, attackers face significant challenges in ensuring that their injected samples are actually collected into the pretraining corpus. This uncertainty is a realistic and well-recognized limitation.
>
> However, we would like to clarify that our definition of realism is framed from the perspective of **attacker-side feasibility**. We do not assume that the attacker has control over the entire data collection pipeline. Instead, we focus on a more common and actionable setting: **Does there exist a plausible data injection path? And if the poisoned samples are eventually collected, can they exert substantial influence on the model?**
>
> From this perspective, we believe our study is grounded in a realistic setting with meaningful security implications:
>
> - Current large-scale multimodal models frequently rely on web-scraped datasets such as LAION and CC3M, which contain a significant amount of open-access content susceptible to manipulation. Prior work [1] has shown that with a budget of just around $1,000, one can successfully poison 0.08% of CC3M and 0.12% of CC12M samples, indicating the **real-world feasibility of low-cost data injection**.
>
> - In contrast, our study requires an even lower poisoning rate, e.g., only 0.004% (for poisoning attacks) and 0.025% (for backdoor attacks), yet achieves **substantial model misalignment** and can bypass **state-of-the-art defense mechanisms**. This further underscores the severity and real-world risk if such attacks succeed.
>
> In summary, although the collection of poisoned samples is inherently uncertain, the **existence of a viable injection path** and the **significant potential consequences** justify the need for serious consideration and systematic study of such attacks. We will incorporate this discussion in the revised version to more clearly articulate our framing of realism and its scope in the context of our threat model.
>
> [1] *Carlini, N., Jagielski, M., et al. “Poisoning web-scale training datasets is practical” S&P (2024).*
>
>
> **R2-Q2&W2:** Thank you for raising the issues of *training time quantification* and *scalability to multi-class attacks*. We provide the following clarifications:
>
> (1) **Training Time**
>
> In the attack preparation phase, we trained the poisoned text generation module (a 6-layer Transformer decoder) for 32 epochs using four RTX 4090 GPUs, with a total training time of approximately 50 hours. The training is based on the ViT-B/32 vision encoder released by OpenAI, which remains frozen throughout the process. Therefore, training is limited to the text-side decoder.
>
> Notably, this training is performed only once during attack preparation and is independent of specific target classes. No retraining is required during subsequent attacks. More importantly, the trained decoder generalizes well and supports the generation of poisoned samples for arbitrary target classes. We reused the same decoder across multiple tasks (including both poisoning and backdoor attacks) without fine-tuning, while consistently achieving strong attack performance. This significantly reduces both the computational cost and operational complexity for the attacker.
>
> As a result, while there is a non-negligible initial training cost, it is highly reusable and does not grow linearly with the number of target classes. This design makes the approach practically feasible and attack-efficient.
>
> (2) **Scalability to Multi-Category Attacks**
>
> Our method is inherently designed to support *parallel poisoning across multiple target classes*, as demonstrated in the following aspects:
>
> - *Efficiency*: In the supplementary experiments, we generated poisoned texts for 24 target classes in parallel, with 10 samples per class. The total generation time was approximately 50 seconds, i.e., around 50 seconds per class and about 5 seconds per sample. Since different classes can be processed in parallel, the overall computational cost does not scale linearly with the number of classes, demonstrating strong scalability.
>
> - *Effectiveness*: The main results presented in the paper (e.g., Table 1) are based on poisoned samples generated under this multi-class setting. The attack success rates under parallel generation were comparable to those in single-target scenarios, indicating that the method maintains strong and stable attack effectiveness in the multi-class setting.
>
> In summary, our approach features a one-time, reusable training process and demonstrates high efficiency and stability in multi-class scenarios, highlighting both its real-world feasibility and threat potential.
>
> **R3-Q3:** Thank you for noting the lack of examples. We agree that visualizations are important for understanding the attack mechanism.
>
> To address this, we will include a set of visualized poisoning text examples in the revised version. These examples illustrate the poisoned samples generated by our method and compare them with those produced by the baseline. The contrast helps reveal differences in background composition, providing clearer insight into how our approach manipulates semantic alignment to mislead model predictions.
>
> To further strengthen the quantitative analysis, we will also report the similarity score (*sim\_score*) between the poisoned text embedding and the centroid embedding $Z_B$ of the corresponding target class. This score reflects the degree of semantic alignment between the background content and the target concept. Our results show that *sim\_score* is consistently higher for samples generated by our method, confirming the effectiveness of background alignment in facilitating successful attacks.
>
> The table below presents representative poisoning examples under different attack methods. In each example, the **target object** is highlighted in bold, *background descriptions* are italicized, and specific $\underline{background~entities}$ are underlined:
> |Target Labels|Attack Methods|Poisoning Texts|*sim\_score*|
> |:-:|:-:|:-:|:-:|
> |refrigerator|Baseline |*casual young $\underline{girl}$ leaning on* **an opened refrigerator.**|75.30|
> |refrigerator|ToxicTextCLIP|**The refrigerator** *is best positioned in the $\underline{kitchen}$, flanked by $\underline{dish cabinets}$ on either side.*|87.72|
> |zebra|Baseline|*$\underline{people}$ watch* **the zebras** *from the deck of the $\underline{tent}$.*|72.36|
> |zebra|ToxicTextCLIP|*A photo of* **zebra, the black and white striped zebra** *on the $\underline{grasslands}$. $\underline{Wildlife}$ photography.*|80.23|
>
>
> **R4-Q4:** Thank you for raising the interaction between text and image attacks. We address this both conceptually and empirically.
>
> Since CLIP consists of both an image encoder and a text encoder, multimodal attacks can be constructed in the following four combinations: **image poisoning + text poisoning**, **image backdoor + text poisoning**, **image backdoor + text backdoor**, and **image poisoning + text backdoor**.
>
> (1) **Image Poisoning + Text Poisoning**
>
> In single-modal image poisoning, two typical strategies exist:
> - *Label-flip poisoning*: only replacing the label of a source image (e.g., “dog”) with that of a target class (e.g., “cat”);
> - *Clean-label poisoning*: embedding features of the source class (e.g., “cat”) into images of the target class (“dog”) while keeping the target label, thereby inducing semantic confusion.
>
> When extended to CLIP, image poisoning must be paired with corresponding text to influence joint embeddings. To simulate this multimodal poisoning scenario, we adopt the MetaPoison method [2] for image-side poisoning and replace the associated text with poisoning samples generated by our ToxicTextCLIP approach. This enables stronger cross-modal misalignment.
>
> As shown below, combining our method with image poisoning leads to a higher attack success rate (ASR) compared to image-only poisoning:
> |Attack Methods|ASR|
> |:----:|:----:|
> |MetaPoison|27.45|
> |MetaPoison+ToxicTextCLIP|35.19|
>
> (2) **Image Backdoor + Text Poisoning**
>
> We further evaluate the interaction between image backdoor attacks and our text poisoning. For the image-side backdoor, we use ImgBackdoor [3]; for the text, we pair poisoned prompts generated by ToxicTextCLIP.
>
> The table below shows that combining these attacks improves top-k retrieval accuracy (Hit@1-Hit@10) relative to using the image backdoor alone, indicating a clear synergistic effect between modalities:
> |Attack Methods| Hit@1|Hit@5|Hit@10|
> |:----:|:----:|:----:|:----:|
> |ImgBackdoor|79.27|90.33|97.74|
> |ImgBackdoor +ToxicTextCLIP|85.73|96.27|99.35|
>
> (3) **Image Backdoor + Text Backdoor**
>
> This configuration has limited practical relevance. It requires simultaneous activation of both visual and textual triggers, which limits stealth, flexibility, and real-world feasibility. As such, we do not consider this a meaningful threat model in practice.
>
> (4) **Image Poisoning + Text Backdoor**
>
> This setting intends to use a text backdoor to activate poisoned image classes, but remains unexplored due to the lack of matching image-side techniques. We therefore leave its evaluation for future work.
>
> In summary, both **image poisoning+text poisoning** and **image backdoor+text poisoning** outperform single-modality attacks, revealing a cross-modal synergy that can be exploited. We will include the corresponding analysis and results in the revised manuscript to better reflect the practical threat.
>
> [2] *Huang, W., Geiping, J. et al. “MetaPoison: Practical General-purpose Clean-label Data Poisoning” NeurIPS (2020).*
>
> [3] *Carlini, N., Terzis, A., et al. “Poisoning and backdooring contrastive learning” ICLR (2022).*

---

> > ### Comment · Reviewer_ezQ3 · 2025-08-05
> >
> > I thank the author for their rebuttal.
> >
> > I would like to raise two points for further consideration. First, while plausible injection paths exist in today's web-based data collection pipelines, **there is no guarantee that CLIP-based models, especially those intended for deployment, will rely on such data sources**. In fact, providers are increasingly expected to use their own curated datasets, as anticipated by regulatory frameworks like the U.S. AI Act and the EU AI Act [1]. This trend may reduce the practical feasibility of poisoning attacks that depend on uncontrolled data ingestion from the web.
> >
> > Second, upon revisiting the paper, it appears that the defense Safeguard (Better Safe Than Sorry: Pre-training CLIP Against Targeted Data Poisoning and Backdoor Attacks)[2] was not included in the comparison. This method has demonstrated stronger performance than several defenses evaluated in the paper, and its inclusion would provide a more complete and up-to-date evaluation of current mitigation strategies.
> >
> > [1] European Union – EU AI Act (Regulation (EU) 2024/1689, in force since August 1, 2024)
> >
> > [2] Yang, Wenhan, Jingdong Gao, and Baharan Mirzasoleiman. "Better safe than sorry: Pre-training clip against targeted data poisoning and backdoor attacks." arXiv preprint arXiv:2310.05862 (2023).

---

> > > ### Author Response · Authors · 2025-08-05
> > > **Response to Reviewer's Comments on Future Threat Relevance and SafeCLIP Defense**
> > >
> > > Thank you very much for your thoughtful and forward-looking comments after carefully reviewing our paper. We have carefully reflected on your suggestions and provide the following response:
> > >
> > > **First**, regarding your observation that “future training data may become more controlled, potentially diminishing the feasibility of open-source data poisoning pathways,” we fully acknowledge the importance of this long-term trend. With the gradual implementation of regulatory frameworks such as the EU AI Act and the U.S. AI Act, model developers are indeed increasingly moving toward internally curated or audit-compliant datasets to enhance data governance and overall security.
> > >
> > > At the same time, we observe that the field as a whole is still in a transitional phase toward more rigorous data governance. During this period, **current mainstream model training practices, including many foundation models and multimodal pretraining systems, still heavily rely on large-scale open image-text datasets such as LAION and CC12M**. These datasets currently lack robust and consistent vetting mechanisms, leaving data poisoning attacks feasible and potentially impactful in real-world settings.
> > >
> > > This study is motivated by such a context: we focus on the injection risks within partially uncontrolled training pipelines and propose an actionable text-based attack strategy. We believe this work holds practical relevance at the current stage and also sheds light on potential attack pathways that may emerge under more regulated data environments. For example, even in scenarios where data sources become more closed, adversaries may still construct semantically aligned yet manipulative “honeypot samples” to mislead data filtering mechanisms and gain influence over model behavior. Therefore, we believe the methodological framework proposed in this study is to some extent transferable and can offer conceptual and methodological support for future defense modeling.
> > >
> > > **Second**, regarding your suggestion to consider the SafeCLIP defense (Yang et al., 2023), we are currently conducting an urgent supplementary experiment to preliminarily evaluate its defensive effectiveness. The corresponding results will be made available shortly via a public system comment. We appreciate your patience and understanding. In addition, we will include a detailed analysis of the method’s mechanism and relevance in our revised version to further enhance the comprehensiveness and timeliness of the defense evaluation section.
> > >
> > > Once again, we sincerely thank you for the depth of thought and professional judgment you demonstrated during the review process. Your feedback has not only helped us improve the current work but also provided valuable inspiration for broadening future research directions.

---

> > > ### Author Response · Authors · 2025-08-07
> > > **Supplementary Results for SafeCLIP (Yang et al., 2023)**
> > >
> > > We sincerely appreciate your valuable suggestion regarding the inclusion of the SafeCLIP defense.
> > >
> > > Following your recommendation, we have conducted a preliminary evaluation comparing SafeCLIP with other state-of-the-art defenses previously assessed in our manuscript. To ensure a systematic and fair assessment, all models were trained for the same number of epochs across the three defense methods. The results are summarized in the table below:
> > >
> > > |Attack Type|Defense|Method|ASR|Hit@1|Hit@5|Hit@10|
> > > |:-:|:-:|:-:|:-:|:-:|:-:|:-:|
> > > |STI-P |RoCLIP|mmPoison|33.33|-|-|-|
> > > |STI-P |RoCLIP|ToxicTextCLIP|**$\underline{70.83}$**|-|-|-|
> > > |STI-P |CleanCLIP|mmPoison|45.83|-|-|-|
> > > |STI-P |CleanCLIP |ToxicTextCLIP|**$\underline{75.00}$**|-|-|-|
> > > |STI-P |SafeCLIP|mmPoison|25.00|-|-|-|
> > > |STI-P |SafeCLIP|ToxicTextCLIP|**$\underline{64.17}$**|-|-|-|
> > > |S-BD |RoCLIP|Baseline|-|57.82|86.63|96.42|
> > > |S-BD |RoCLIP|ToxicTextCLIP|-|**$\underline{91.15}$**|**$\underline{99.25}$**|**$\underline{99.62}$**|
> > > |S-BD |CleanCLIP| Baseline |-|56.29|88.58|95.48|
> > > |S-BD |CleanCLIP |ToxicTextCLIP|-|**$\underline{86.63}$**|**$\underline{99.44}$**|**$\underline{100.00}$**|
> > > |S-BD |SafeCLIP| Baseline |-|20.17|33.02|45.69|
> > > |S-BD |SafeCLIP|ToxicTextCLIP|-|**$\underline{60.96}$**|**$\underline{70.28}$**|**$\underline{82.53}$**|
> > >
> > > The experimental results show that ToxicTextCLIP significantly outperforms baseline attacks under all defense settings. Specifically, in the STI-P task, ToxicTextCLIP achieves substantially higher attack success rates (ASR) compared to mmPoison. For example, under SafeCLIP, ASR increased substantially from 25.00% to 64.17%. Similarly, in the S-BD scenario, ToxicTextCLIP exhibits remarkable improvements in Hit@k metrics, for instance, increasing the Hit@1 metric from 20.17% to 60.96% under the SafeCLIP defense. These results indicate that ToxicTextCLIP maintains strong attack capabilities across multiple defense scenarios.
> > >
> > > The main reason behind the above results lies in the three-stage design of SafeCLIP. First, it performs single-modal warmup to help the model adapt to modality-specific features. Second, it applies multi-modal contrastive training to strengthen semantic alignment between text and images. Third, it uses a GMM model to divide the data into “safe” and “malicious” subsets, thereby filtering out potentially abnormal samples during training. However, the adversarial texts generated by ToxicTextCLIP are semantically natural and exhibit highly aligned background content. This significantly increases the cosine similarity of poisoned pairs during SafeCLIP’s contrastive learning phase, making them more likely to pass the GMM-based “safe subset” filtering and be included in the contrastive training process. In contrast, methods like RoCLIP and CleanCLIP lack such filtering mechanisms and are thus more vulnerable to this type of semantic-level attack. These findings further reveal the limitations of existing defenses in addressing semantic manipulation, highlighting the need for future work on more semantically aware defense frameworks.
> > >
> > > We fully agree that incorporating SafeCLIP into our evaluation greatly improves the completeness and timeliness of our study. Accordingly, we will formally include these comparisons in the revised manuscript.
> > >
> > > Once again, thank you very much for your detailed and insightful comments, which have substantially helped us enhance both the depth and clarity of our paper.

---

> > > > ### Comment · Reviewer_ezQ3 · 2025-08-07
> > > >
> > > > I thank the authors for their detailed follow-up and clarifications. I will adjust my score accordingly.

---

> > > > > ### Author Response · Authors · 2025-08-07
> > > > > **Thank You for Your Feedback**
> > > > >
> > > > > Thank you for your follow-up and for taking the time to reconsider our work. We truly appreciate your feedback. We're grateful for your time and thoughtful input.

---

### Note · Authors · 2025-08-12

This work studies text-side poisoning during CLIP pre-training and proposes *ToxicTextCLIP*, which injects background-consistent prompts to systematically disrupt vision-language alignment. Reviewers noted innovation and practical value in the problem formulation, method design, and attack effectiveness, while requesting further clarification on method applicability and defense evaluation. In the rebuttal, we responded with additional experiments, ablations, efficiency and naturalness assessments, and comparative evaluations against defenses.

The main discussion points are:
- **Adequacy of theoretical grounding**: Our method is not an empirical assembly but an architecture explicitly grounded in the background alignment hypothesis. The class embedding center ($Z\_B$) forms a stable semantic anchor via prompt averaging, and the query matrix ($Q$) drives fine-grained interactions between tokens and image regions to dynamically steer semantics during generation. These components are tightly coupled to the attack objective and, in ablations, are validated as indispensable modules.
- **Whether text-side attacks are redundant**: We demonstrate the independence and mechanistic differences of text poisoning. Text directly influences the learning trajectory of semantic representations and, early in training, misguides contrastive pairs through background consistency. Cross-modal experiments show synergistic gains with image-side triggers, indicating complementarity rather than redundant stacking.
- **Sufficiency of defense evaluation**: We augmented evaluation with representative multimodal defenses (e.g., SafeCLIP) and text-specific defenses (e.g., ONION), and reported metrics on textual naturalness, semantic consistency, and scalability to cover key attack/defense dimensions. The results suggest that future defenses should emphasize deeper semantic-consistency modeling.

**All reviewers who participated in the discussion indicated that their concerns were resolved; no new objections were raised during the rebuttal phase, and there are currently no outstanding technical issues.**

---

### Decision · Program_Chairs · 2025-09-17

**Decision:**

Accept (poster)

**Comment:**

Initially the reviews were mixed but, after the rebuttal and discussion phase, they all recommend acceptance.